# Predicting Parkinson's disease trajectory using clinical and functional MRI features: A reproduction and replication study

Elodie Germani[1]*, Nikhil Bhagwat[2], Mathieu Dugré[3], Rémi Gau[2], Albert A. Montillo[4], Kevin P. Nguyen[4], Andrzej Sokolowski[3], Madeleine Sharp[2], Jean-Baptiste Poline[2☯], Tristan Glatard[3☯]

**1** Univ Rennes, Inria, CNRS, Inserm, Rennes, France, **2** Department of Neurology and Neurosurgery, McGill University, Montreal, Canada, **3** Department of Computer Science and Software Engineering, Concordia University, Montreal, Canada, **4** Lyda Hill Department of Bioinformatics, University of Texas Southwestern Medical Center, Dallas, TX, United States of America

☯ These authors contributed equally to this work.
* elodie.germani@irisa.fr

**Data Availability Statement:** Data used in the preparation of this article were obtained on August 21st, 2023 from the Parkinson's Progression Markers Initiative (PPMI) database

## Abstract

Parkinson's disease (PD) is a common neurodegenerative disorder with a poorly understood physiopathology and no established biomarkers for the diagnosis of early stages and for prediction of disease progression. Several neuroimaging biomarkers have been studied recently, but these are susceptible to several sources of variability related for instance to cohort selection or image analysis. In this context, an evaluation of the robustness of such biomarkers to variations in the data processing workflow is essential. This study is part of a larger project investigating the replicability of potential neuroimaging biomarkers of PD. Here, we attempt to fully reproduce (reimplementing the experiments with the same methods, including data collection from the same database) and replicate (different data and/or method) the models described in (Nguyen et al., 2021) to predict individual's PD current state and progression using demographic, clinical and neuroimaging features (fALFF and ReHo extracted from resting-state fMRI). We use the Parkinson's Progression Markers Initiative dataset (PPMI, ppmi-info.org), as in (Nguyen et al., 2021) and aim to reproduce the original cohort, imaging features and machine learning models as closely as possible using the information available in the paper and the code. We also investigated methodological variations in cohort selection, feature extraction pipelines and sets of input features. Different criteria were used to evaluate the reproduction attempt and compare the results with the original ones. Notably, we obtained significantly better than chance performance using the analysis pipeline closest to that in the original study ($R2 > 0$), which is consistent with its findings. In addition, we performed a partial reproduction using derived data provided by the authors of the original study, and we obtained results that were close to the original ones. The challenges encountered while attempting to reproduce (fully and partially) and replicating the original work are likely explained by the complexity of neuroimaging studies, in particular in clinical settings. We provide recommendations to further facilitate the reproducibility of such studies in the future.

(www.ppmi-info.org/access-dataspecimens/download-data), RRID:SCR 006431. For up-to-date information on the study, visit www.ppmi-info.org. All data used in this study, as well as a data dictionary, are free and publicly available at the PPMI website, upon an online application, the signature of the Data User Agreement and of the publications policies. For any questions on the PPMI Intellectual Property (IP) Policy or applying for an exception to the PPMI IP Policy to ppmi@michaeljfox.org. To access and download the minimal datasets used in this study, please refer to the notebook available at https://github.com/elodiegermani/nguyen-etal-2021/blob/main/nguyen-etal-2021.ipynb. By running the first cells of the notebooks, until "Download data", you can reproduce the full selection process that we applied to obtain the subset of data from PPMI explored in this study. By using a homemade Python package, this code will allow you to directly download the raw fMRI and MRI data used in this study, along with the associated metadata. ID and password associated to your online application to the PPMI website will be asked during before downloading. Reproduction of all processing and analysis steps of the study can also be done using this notebook. The list of participants, raw and derived data can also be directly send upon request to corresponding authors (elodiegermani@gmail.com), upon an online application to the PPMI website, the signature of the Data User Agreement and of the publication policies. We declare that we did not have any special access privileges that others would not have when attempting to access the data from PPMI.

**Funding:** This work was funded by the Michael J. Fox Foundation for Parkinson's Research (MJFF-021134). This work was also funded by a MITACS Global Research Award (IT34055). This work was partially funded by Region Bretagne (ARED MAPIS) and Agence Nationale pour la Recherche for the programm of doctoral contracts in artificial intelligence (project ANR-20-THIA-0018). The funders had no role in study design, data collection and analysis, decision to publish, or preparation of the manuscript.

**Competing interests:** The authors have declared that no competing interests exist.

## Introduction

Parkinson's disease (PD) is the second most common neurodegenerative disorder with more than 10 million people affected in the world. Disease manifestations are heterogeneous and their evolution varies between patients, dividing them in different subtypes and stages [1]. Identification of these stages is essential for clinical trials as well as for clinical practice to track the disease progression. However, there is currently no established biomarker of disease severity or progression [2, 3].

Neuroimaging techniques are able to capture rich and descriptive information about brain structure and functional architecture non-invasively. In conjunction with computational algorithms based on pattern recognition and machine learning, neuroimaging measures began to emerge as candidate PD biomarkers in the past few years. Among other imaging modalities, functional Magnetic Resonance Imaging (fMRI), which estimates the blood oxygenation level-dependent (BOLD) effect to represent neural activity, showed a high potential in identifying specific biomarkers related to PD and its progression [4]. While disease phenotypes are heterogeneous, neuronal dysfunction patterns were shown to be highly replicable between patients. In [5], authors showed that while the location of the dysfunction within brain networks might vary between individuals, the progression of this dysfunction over time, associated with the progression of the disease itself, was shown to be highly similar between individuals.

Resting-state fMRI (rs-fMRI) features are particularly promising. Region-wise measurements such as regional homogeneity (ReHo) and Amplitude of Low Frequency Fluctuations (ALFF) were used in multiple studies to predict PD trajectory or motor subtypes [6–11]. ReHo quantifies the connectivity between a voxel and its nearest neighboring voxels and was shown to be affected by neurodegenerative diseases [12]. ALFF and its normalized form, fractional ALFF (fALFF), measure the power of the low frequency signals at rest, which mostly consists in spontaneous neuronal activity [13]. In previous studies, specific regional values of these two measures (e.g. ReHo in the putamen and cerebellum, and fALFF in the right cerebellum) were found to be correlated positively or negatively with MDS-UPDRS scores. These findings have been attributed to the role of several brain networks involving these regions in motor function.

However, despite their potential, neuroimaging measures are sensitive to multiple sources of variability that impact their replicability and may explain why the derived biomarkers are not well established in clinical and research practice. In particular, neuroimaging analyses require specific methodological choices at various computational steps, related to the software tools, the method, and the parameters to use. These choices, also known as "researchers' degrees of freedom" [14], might have a large impact on the results of an experiment as they can impact the predictiveness of the signal extracted and can lead to a lack of agreement when analyzing the same neuroimaging dataset with different analysis pipelines [15, 16]. For instance, in task-based fMRI, 70 research teams were asked to analyze the same fMRI dataset using their usual analysis pipeline and results were substantially variable across teams [16].

Furthermore, neuroimaging results have been shown to be impacted by differences in hardware architectures or software package versions [17, 18], questioning the robustness of the results. This suggests that a single pipeline evaluation is not sufficient to obtain robust results. A poor robustness of the results would question their reliability, since significant results might have been obtained by chance and might actually be false positive findings [19]. This robustness can be assessed by studying the distribution of results across perturbations of the workflow.

There are also concerns about the reproducibility of machine learning studies. Indeed, in a recent study [20], researchers attempted to reproduce several machine learning experiments, revealing multiple issues which could lead to the non-reproducibility of findings. These issues

can be split in three categories [21]: data leakage, computational reproducibility, and choice of evaluation metrics. In particular, [22] performed a review of CNN-based classification of Alzheimer's subtypes and found a potential data leakage in half of the 32 surveyed studies due to a wrong data split at the subject-level, a data split after data augmentation or dimension reduction, transfer learning with models pre-trained on parts of the test set or the absence of an independent test set. Such a data leakage, which we did not notice in our study, might cause an over-optimistic performance assessment of models and thus, a lack of reproducibility and replicability of the findings. Evaluation procedures can also cause the non-reproducibility of findings, due to unsuitable metric choices when using unbalanced datasets for instance or questionable cross-validation procedures, in particular with low sample sizes. Random choices in a training procedure, for instance initial weights or hyper-parameters random selection, which all impact computational reproducibility, might also lead to uncontrolled fluctuations in results when using different random initialization states.

Conflicting terminologies exist for the terms reproducibility and replicability [23]. Here, we define reproducibility as attempts made with the same methods and materials. The aforementioned method can include a data collection step, and in that case, we consider that the same materials correspond to the original database where data collection is performed. Replicability, on the other hand, is tested with different but comparable materials or methods, assuming that the tested pipelines are all suitable to extract signal from the data. Note that the term comparable is ambiguous, but we define its use in the context of this study in the Method Section.

Replicability experiments have shown different degrees of variability between findings obtained with different analytic conditions. These studies are usually done using healthy populations, as in [16]. For clinically-oriented research, i.e. using patient populations, however, the topic remains understudied. Such studies requires a specific attention as they are useful to develop new biomarkers that can influence treatment development and clinical trial applications. These studies also often target specific populations of patients with unique characteristics, in particular for PD for which inter-individual variability is high [24]. Such studies often use small sample sizes, which has been shown to lead to a lower reproducibility and replicability of findings [25, 26]. Reproducibility and replicability of studies in clinical settings is of higher importance to improve the trustworthiness of new biomarkers, which is an important factor that would facilitate their development and application in clinical practice..

In this paper, we evaluate the reproducibility and replicability of the study in [6], a clinically-oriented study on a PD population. The study in [6] is of particular interest as it uses the Parkinson's Progression Markers Initiative (PPMI) dataset [27], a large open access dataset to study Parkinson's disease. Moreover, it investigates the clinically relevant problem of trying to predict an individual's current and future disease severity over up to 4 years and it uses two different rs-fMRI-derived biomarkers: ReHo and fALFF. In [6], the authors, including current co-authors KPN and AAM, trained several machine learning models using regional measurements of ReHo or fALFF along with clinical and demographic features to predict Movement Disorder Society-Unified Parkinson's Disease Rating Scale (MDS-UPDRS) total score at acquisition time and up to 4 years after. They selected n = 82 PD patients by searching for all patients available at that time with rs-fMRI and MDS-UPDRS score at the same visit from the PPMI database and preprocessed the functional images to extract whole-brain maps of fALFF and ReHo. They compared three atlases, splitting the brains in different numbers of regions to extract mean region-wise features which are fed to the machine learning models. They achieved better than chance performance for prediction at each time point with both fALFF and ReHo, e.g. r-squared of 0.304 and 0.242 for prediction of current severity with ReHo and fALFF respectively. Finally, the authors discussed the most important brain regions for prediction. Although most studies do not perform external validation, authors of [6] confirmed the

predictiveness of their models on an external dataset, the next largest dataset available at the time: the Parkinson's Disease Biomarkers Program (PDBP) from NIH. On this dataset, they found reproducible model performance.

Different criteria could be used to conclude on success of the reproduction and replication of this study: 1) if the models trained on fALFF and ReHo at each time points showed better than chance performance in terms of r-squared (R2 >0 and R2 >chance-model R2) when tested on the PPMI dataset using the evaluation procedure proposed in [6] and 2) if these models showed similar performance (R2 greater than 0 and absolute difference between original and reproduction R2 less than 0.15) to those proposed in the original study. Our main interests were to assess the difficulties and challenges of reproducing fMRI research experiments, thus we first attempted to reproduce the study without contacting the authors to assess the importance of publicly-shared resources and description given in the paper. After that, we contacted the authors to better understand the failure of our initial reproduction attempt.. But our goal was also to further evaluate the impact of different analytical choices (*e.g.* processing pipeline, choice of feature set, etc.) on the results of these experiments. In this paper, we explore how these choices affect different parts of the analysis:

- Cohort selection and sample size,

- fMRI pre-processing pipeline,

- fMRI feature quantification,

- Choice of input features for machine learning models,

- Machine learning models choice and results reporting.

A primary purpose of this investigation is also to learn about the difficulties encountered to reproduce neuroimaging studies, in particular in clinical research settings, and to provide some recommendations on best practices to facilitate the reproducibility of such studies in the future. This study is part of a larger effort to explore the reproducibility and robustness of PD biomarkers extracted from neuroimaging data, but this study is the first to explore the robustness of fMRI related biomarkers of PD.

## Materials and methods

Our study consisted of two steps:

- Phase 1: a first reproduction attempt without contacting the authors, using only publicly-shared resources available with the original paper.

- Phase 2: a second reproduction attempt after contacting the authors, to obtain more accurate information on the original study.

This two-step reproduction was meant to assess the challenges of reproducing a study using only publicly available materials and to evaluate the contribution of data and code sharing platforms to results reproducibility. In Phase 1, since the materials used by the authors in the original study were not all publicly available, we were not able to make a proper reproduction attempt of the study, we will thus refer to the attempts made at this step as "replications", and to the attempt made after contacting the authors (Phase 2) and using original materials and method as "reproduction". In Phase 2, we performed two different reproduction attempts: a full reproduction, including the cohort selection and data preprocessing steps, and a partial reproduction where we used the derived data provided by the original authors.

## Dataset

As in the original study, we used data available from the Parkinson's Progression Markers Initiative (PPMI) dataset [27], a robust open-access database providing a large variety of clinical, imaging data and biologic samples to identify biomarkers of PD progression. The PPMI study was conducted in accordance with the Declaration of Helsinki and the Good Clinical Practice (GCP) guidelines after approval of the local ethics committees of the participating sites. We signed the Data Use Agreement and submitted an online application to access the data. More information about study design, participant recruitment and assessment methods can be found in [27]. We note that access to such data does not permit us to share such data on our own. Moreover, unlike code repositories with version control numbering, most data repositories are not version controlled, making re-retrieval of data years later thorny.

## Summary of experiments

Reproducing an analysis can be challenging due to (1) the lack of specific information on analysis pipelines, software versions, or specific parameter values, (2) the presence of confusing terms in the available information, (3) the evolution of the software and data materials used in the original study. Our study consisted of 5 global steps: cohort selection, image pre-processing, imaging features computation, choice of input features and model choice and reporting. We used the information available in the original paper and for some parts of the analysis, we also had access to the code shared by the authors on GitHub (*e.g.* for feature computation and machine learning models). Though the authors also made their contact information plainly available, in Phase 1, we wished to work independently of any author contact. Under this scenario, we had to make informed guesses due to the 3 types of challenges stated above, which resulted in a high number of possible workflows. To evaluate the effect of each variation at each step, we defined a *default replication workflow* to which each variation was compared to. At each step, if a variation of the workflow was tested, the other steps were implemented as in the default one. This default workflow was the most likely according to the code shared along with the paper. Fig 1 summarizes the different variations tested and the *default workflow*.

## Cohort selection

The cohort reported in [6] was composed of the largest set of PPMI available at the time, and consisted in 82 PD participants with rs-fMRI and MDS-UPDRS scores obtained during the same visit. MDS-UPDRS Part III (motor examination) was conducted when patients were under the effect of PD medication. Of these 82 participants, 53 participants also had MDS-UPDRS scores available at Year 1 after imaging, 45 at Year 2, and 33 at Year 4.

**Replication cohort.** In Phase 1, we first attempted to reproduce the cohort selection process of [6] using only the information available in the code shared on GitHub and the paper. Based on this information, we filtered the PPMI database using 4 criteria:

- Participants belong to the "Parkinson's disease" cohort, as defined in PPMI.

- Participants have an fMRI acquisition and a MDS-UPDRS score, with MDS-UPDRS Part III conducted ON-medication ("PAG_NAME" different from "NUPDRS3" in the PPMI score file) computed at the same visit (same visit code in PPMI database). Thus, only participants with valid values for MDS-UPDRS Part III score were included in the cohort.

- Participants and visits were also filtered depending on the type of fMRI acquisition. We queried the database with the exact same information as in the S1 Table of the original paper

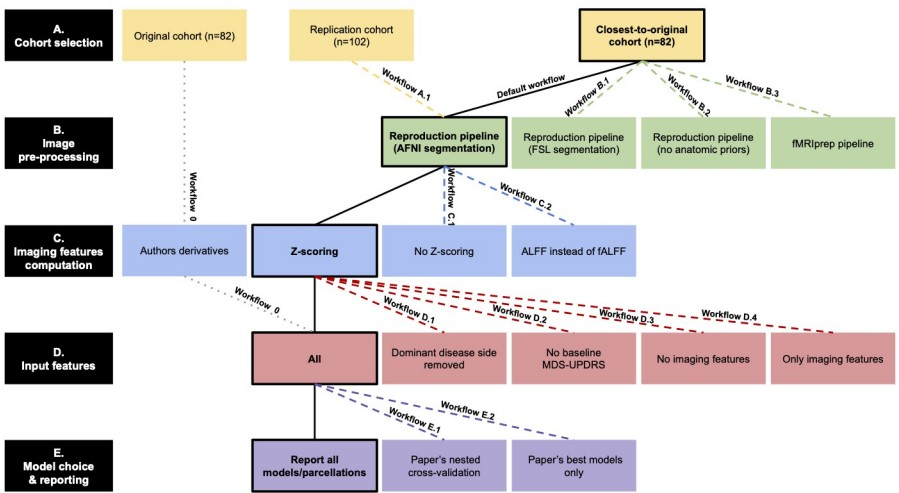

**Fig 1. Summary of the different workflows implemented to attempt to reproduce and replicate the results of** [6] **and explore their robustness to different analytic conditions.** Bold and bordered cells represent the implementation of the default replication workflow at each step, this whole workflow is labeled *Default workflow* and is represented using a plain bold line. The different replication workflows (Phase 1) are represented in dashed lines: all steps different from the variation follow the default workflow and each workflow corresponds to one variation from the default one. The partial reproduction attempt obtained with derived data provided by authors of [6] (Phase 2) is represented with a point-style line.

- *Workflow 0*—partial reproduction using authors derivatives.

**Replication with variations of cohort selection (A):**
- *Workflow A.1*—default workflow with replication cohort.

**Replication with variations of pre-processing pipeline (B):**
- *Workflow B.1*—default workflow with FSL segmentation,
- *Workflow B.2*—default workflow without structural priors,
- *Workflow B.3*—fMRIprep pipeline.

**Replication with variations of feature computation (C):**
- *Workflow C.1*—default workflow with no Z-scoring,
- *Workflow C.2*—default workflow with ALFF.

**Replication with variations of input features (D):**
- *Workflow D.1*—default workflow with no dominant disease side,
- *Workflow D.2*—default workflow with no Baseline MDS-UPDRS,
- *Workflow D.3*—default workflow with no imaging features,
- *Workflow D.4*—default workflow with only imaging features.

**Replication with variations in model choice and reporting (E):**
- *Workflow E.1*—default workflow with paper's nested cross-validation,
- *Workflow E.2*—default workflow with only paper's best model reporting.

(field strength = 3T, scanner manufacturer = Siemens, pulse sequence = 2D EPI, TR = 2400ms, TE = 25ms).

- We also filtered the database to keep only participants for which the visit date and archive date of the image was set before January 1st, 2020 (more than a year before the original study publication) since without contacting the authors we had somewhat imprecise information about the date the authors accessed the database. Note that the choice of this date was made to reproduce as closely as possible the condition of the original database filtering, but other filters could have been used.

This query involved both fMRI metadata obtained using a utility functions from the Python packages livingpark-utils v0.9.3 and ppmi_downloader v0.7.4 and the MDS-UPDRS-III file from the PPMI database.

Since the PPMI database does not permit querying the database at any prior time point, we queried the database at the then current time. Specifically, we queried the PPMI database on August 21st, 2023 and we included the participants selected using these filters in the Baseline time point of our replication cohort. To find the participants who also had a score available at Year 1, Year 2, or Year 4 follow-up, we looked for the visit date associated with the MDS-UPDRS score at Baseline and searched for participants that also had a score at 365 days (1 year) +/- 60 days (2 months), $2 \times 365$ days (2 years) +/- 60 days (2 months) and $4 \times 365$ days (4 years) +/- 60 days (2 months). This method was also used by the original authors to search for their cohort at Year 1, Year 2, and Year 4 follow-up.

**Closest-to-original cohort.** During Phase 2, after contacting the authors (KPN and AAM), the exact participant and visit list used at Baseline was provided to us. We queried the PPMI database using this list and compared with our replication cohort.

The 82 participants of the original Baseline cohort were all included in our replication cohort. For 4 of them, the visit used in our replication cohort was different from the one used in the original cohort. For two participants, we used an earlier visit than the authors: V06 (2 years) instead of V10 (4 years) and BL (baseline) instead of V04 (1 year). For the last two participants that had different visits selected in the replication cohort, images of the visits used by the original authors were not available in the PPMI database when we queried it. We assumed that this issue resulted from the update of the PPMI database in September 2021, and that there is no way to query prior versions of the database, and that the original authors are not allowed to share the original images they obtained when they accessed the database.

The 82 participants of the original cohort that were also included in our replication cohort were used to build a "closest-to-original" cohort to compare with the original cohort. The authors also provided the participant identifiers included at Year 1, Year 2 and Year 4, but we did not have the exact visit used at these time points. Thus, for each time point, we searched for the participants involved in our replication cohort for this time point that were in the list provided by the authors. Several participants from the list provided by the authors were not found in our cohorts. When checking the UPDRS-III files for these missing participants, we found the potential visit used by the authors, but these did not meet the criteria set to select the valid UPDRS-III scores (i.e. "PAG_NAME" was equal to "NUPDRS3" for these visits, but these were discarded when selecting only ON medication scores). For one participant missing in the Year 2 time point, we have not found any visit 2 years +/- 2 months after the Baseline visit. The visit selected for this participant was different in our cohort compared to the original authors cohort due to missing images, which could explain the reason for not finding back this participant for the Year 2 time point. Table 1 summarizes the cohort selection process. This "closest-to-original" cohort corresponds to the one obtained in Phase 2, with a full reproduction attempt of the experiments: the same data collection strategy as the original authors and

**Table 1. Summary of cohort selection procedure.** PPMI global query corresponds to the replication cohort, highlighted in **Cyan**. Participants belonging to the list provided by the authors composed the closest-to-original cohort, highlighted in **Green**.

| Criterions | N |
|---|---|
| **PPMI global query—Baseline** | **102** |
| Participants belonging to the list provided by the authors at Baseline | **82** |
| Participants not belonging to the corresponding session list | 4 |
| Original session after the one obtained with PPMI query | 2 |
| Image of original session not available anymore in PPMI | 2 |
| **PPMI global query—Year 1** | **67** |
| Participants belonging to the list provided by the authors at Year 1 | **51** |
| Participants not belonging to original list | 2 |
| PAG_NAME was NUPDRS3 | 2 |
| **PPMI global query—Year 2** | **61** |
| Participants belonging to the list provided by the authors at Year 2 | **41** |
| Participants not belonging to original list | 4 |
| PAG_NAME was NUPDRS3 | 3 |
| Absence of corresponding score at follow-up time point | 1 |
| **PPMI global query—Year 4** | **46** |
| Participants belonging to the list provided by the authors at Year 4 | **30** |
| Participants not belonging to original list | 3 |
| PAG_NAME was NUPDRS3 | 3 |

the list of participants and sessions of the different cohorts provided by the original authors. Note that even using the same data collection strategy and the participants list, retrieving the same data was not possible due to the changes in the PPMI database mentioned above.

## Image pre-processing

We downloaded functional images from the PPMI database manually for all participants selected in the replication cohort by using the image identifiers corresponding to the participants and visits selected. We also downloaded T1w images corresponding to the participants and visits selected in the replication cohort. If multiple T1w images were available for a participant at a given visit, we selected the one with the smallest identifier number (1st one in the meta-data table). Imaging data from the PPMI online database were available in DICOM format. We converted them into the NIfTI format and we reorganized the dataset to follow the Brain Imaging Data Structure (BIDS) [28] (RRID:SCR_016124) using HeuDiConv v0.13.1 [29] (RRID:SCR_017427) on Docker v20.10.16.

**Default pipeline.** In Phase 1, to pre-process the data, we tried to build a pipeline reproducing the one described by the authors in [6] without contacting them for any additional information or code (which has since been provided). The paper mentions that fMRI images were first realigned to the mean volume with affine transformations to correct for inter-volume head motion, using the MCFLIRT tool in the FSL toolbox [30] (RRID:SCR_002823). Then, images were brain-masked using AFNI 3dAutomask [31] (RRID:SCR_005927). Non-linear registration was performed directly to a common EPI template in MNI space using the Symmetric Normalization algorithm in ANTS [32] (RRID:SCR_004757). For denoising, motion-related regressors computed using ICA-AROMA [33] were concatenated with the nuisance regressors from affine head motion parameters computed with MCFLIRT and mean timeseries of white matter and cerebrospinal fluid. These nuisance signals were regressed out

of the fMRI data in one step (i.e. all confounds concatenated in a single matrix and regressed from voxels timeseries).

Using this information, we built the closest-possible pipeline to this description. More details will be given below regarding the choices that we made to build this pipeline. We implemented this pipeline—referred to as the *default workflow*—using Nipype v1.8.6 (RRID: SCR_002502) [34], FSL v6.0.6.1, AFNI v23.3.01 and ANTs v2.3.4. We executed the pipeline with a custom-built Docker image available on Dockerhub https://hub.docker.com/repository/docker/elodiegermani/nguyen-etal-2021/general and built using NeuroDocker [35] with base image fedora:36 and a miniconda v23.5.2–0 [36] environment with Python v3.10. All pre-processing, feature computation and model training were run using homemade Boutiques descriptors using Docker v20.10.16 and Boutiques v0.5.25 [37]. Boutiques descriptors for image processing and model training are available in Zenodo [38, 39].

In this *default workflow*, functional images were first realigned to the middle volume using FSL MCFLIRT, using affine registration (6 degrees of freedom), b-spline interpolation and mutual information cost function. The motion-corrected images were then skull-stripped using AFNI 3dAutomask with default parameters (clip level fraction of 0.5). Following this, ANTs symmetric normalization algorithm was used to normalize images to the MNI template. First, rigid, affine, and symmetric normalization transformations from native to MNI space were computed using the first volume of the brain-extracted functional images as source image and the MNI152NLin6Asym template, with a 2mm resolution as reference. The exact MNI template used for registration was not mentioned in the original paper. The choice of this particular template for our pipeline was due to the use of ICA-AROMA after registration. Indeed, to run ICA-AROMA in the MNI space or without FSL registration transform matrices, images must be in FSL's default MNI space, which is the MNI152NLin6Asym [40]. We downloaded this EPI template from C-PAC: https://github.com/FCP-INDI/C-PAC/blob/main/CPAC/resources/templates. We applied the computed transformations to functional images using ANTs also with B-Spline non linear registration.

For denoising, we regressed out several nuisance signals from the fMRI data, as in the original study. The 6 affine motion parameters computed using MCFLIRT were used as regressors. In addition, we ran ICA-AROMA v0.4.3-beta on data already registered in MNI space to extract motion-related components. All the components classified as motion-related were added as regressors to each participants.

For white-matter (WM) and cerebrospinal fluids (CSF) signals, there was no information about the method used by the authors to compute these signals in the original paper. Thus, we implemented three different methods to build the default workflow but also to compare the impact of pre-processing pipelines on the results of the study. In the *default workflow*, we arbitrarly chose to use AFNI to compute these regressors. We used the structural T1w images downloaded from PPMI and ran several analysis steps: brain extraction using 3dSkullstrip, segmentation using 3dSeg with defaults parameters, 3dCalc to extract the mask for WM and CSF, 3dResample to resample the masks to the functional image using nearest-neighbors interpolation and 3dMaskave to extract timeseries of voxels inside the WM and CSF masks. Then, we computed the mean timeseries across these voxels for WM and CSF and added these signals as nuisance regressors.

**Variations of the default workflow.** As mentioned in the previous section, we had to make guesses to build the pipeline for the default workflow. For some of these guesses, other valid alternatives would have been possible. In particular, for the extraction of WM and CSF, which could have been made with another software package and/or method. Thus, we also compared this workflow with two other methods to extract WM and CSF signals. The first method (pipeline *B.1—default workflow with FSL segmentation*) used tools from FSL instead of

AFNI to extract structural-derived masks. In this pipeline, BET was used to remove non-brain tissues from structural images, then the images were segmented using FAST to extract WM and CSF masks. The masks were resampled to functional images using affine registration implemented in FLIRT, and mean timeseries inside each mask were extracted using FSL's ImageMeants function in Nipype.

The second method (pipeline *B.2—default workflow without structural priors*) did not involve image segmentation. We used mask templates available in FSL and Nilearn: MNI152_T1_2mm_VentricleMask from FSL for CSF, and WM brain-mask in MNI152 template resolution 2mm in Nilearn v0.10.2 [41] (RRID:SCR_001362) for WM. The masks were resampled to the functional images using a nearest neighbors interpolation in Nilearn, and mean timeseries inside each mask were also computed using Nilearn.

In all pipelines, the nuisance signals were regressed from the functional images in MNI space using FSL RegFilt. The denoised images were then used to compute the imaging features passed as input to the machine learning models.

**Other pipelines variations.** To explore the robustness of the original results to variations in the workflow, we also analyzed the functional and structural images using fMRIprep v23.0.2 [42] (RRID:SCR_016216), a robust pre-processing pipeline that requires minimal user input, and which implements pre-processing steps that are different from the ones used in the default workflow and its variations. This allowed us to see how impactful the changes in image pre-processing pipelines could be in this study. We used default parameters for fMRIprep, except for the reference template that we set to MNI152NLin6Asym with a resolution of 2mm to be able to run ICA-AROMA afterwards [40].

Final preprocessed functional images in MNI space were then passed as input to ICA-AROMA to obtain motion-related components. The 6 motion regressors, WM and CSF mean timeseries extracted by fMRIprep were concatenated to the timeseries of the motion-related components identified by ICA-AROMA and regressed out from the pre-processed images using FSL RegFilt, as in the default workflow. This pipeline is referred to as *B.3—fmriprep pipeline*.

**Quality control.** We implemented quality control checks at different steps of each pipeline. The purpose of these controls was to explore quality of data, but we did not exclude any participant due to data low quality, as this step was not performed in the original paper.

For each participant, we controlled the quality of functional pre-processing (motion correction, brain masking, and registration to MNI space) by superposing the pre-processed functional volume at each time point to an MNI-space brain mask, and visually inspecting a predefined image slice for incorrect registration or masking. We also visually inspected the 6 motion parameters identified during motion correction (rotation and translation in the x, y and z directions). We also computed the frame-wise displacement (FD) of head position as done in [43], calculated as the sum of the absolute volume-to-volume values of the 6 translational and rotational motion parameters converted to displacements on a 50 mm sphere (multiplied by $2 \times \pi \times 50$). We explored these values using the threshold used in [44] for the lenient strategy: identification of participants with mean FD > 0.55mm. Segmentations masks for WM and CSF obtained with the 2 different workflow variations were also visually inspected for failed segmentations. For the fMRIprep pipeline, we validated the quality of the processing using the log files produced by the pipeline, since these produce the same outputs as the quality control steps mentioned above.

## Imaging features computation

**Whole-brain maps computation.** In the original study, mean regional values of z-scored fALFF and ReHo maps were used as input features to the machine learning models, in

addition to several clinical and demographic features. fALFF and ReHo were computed on the denoised fMRI data using C-PAC [45] (RRID:SCR_000862). Voxel-wise ReHo was computed using Kendall's coefficient of concordance between each voxel and its 27-voxel neighborhood. For ALFF and fALFF, linear de-trending and band-pass filtering were first applied to each voxel at 0.01–0.1 Hz, then the standard deviation of the signal was computed to obtain ALFF whole-brain maps. These maps were divided by the standard deviation of the unfiltered signal to obtain whole-brain fALFF maps. Z-scores maps for ReHo and fALFF were calculated at the participant-level.

For each workflow, we used the original code used by the authors, available at https://github.com/DeepLearningForPrecisionHealthLab/Parkinson-Severity-rsfMRI/blob/master/ppmiutils/rsfmri.py. We followed the exact same steps as in the original paper to compute the raw ReHo and fALFF maps. However, a mask file was needed in the authors' code to compute the features. We thus applied AFNI 3dAutomask on the denoised fMRI data to obtain a brain mask for each participant.

The initial code shared by the authors did not include any z-scoring of the whole-brain maps for fALFF and ReHo, thus we used FSL's ImageMaths function to compute the z-score maps. Non z-scored maps (*C.1—default workflow with no Z-scoring*) were also saved and set as input to the models for comparison. We also considered ALFF instead of fALFF as input measure (*C.2—default workflow with ALFF*) as the authors also mentioned having tested this feature. We note that for the second step of the experiment (i.e. after a reproduction attempt using only publicly-available materials, by contacting the authors), the authors of [6] have supplied us with all derived maps.

**Regional features extraction.** In the original paper, regional features were extracted from the ReHo and fALFF whole-brain maps using three different parcellations. These included the 100-ROI Schaefer [46] functional brain parcellation, modified with an additional 35 striatal and cerebellar ROIs, and the 197-ROI and 444-ROI versions of the Bootstrap Analysis of Stable Clusters (BASC) atlas [47]. These parcellations were used to compute the mean regional ReHo or fALFF values for each participant and performance of the machine learning models were compared between the parcellations. For our first attempts at re-implementing the workflow, we did not have access to the modified version of the Schaefer atlas used by the original authors. Thus, we derived a similar custom atlas by using the 100-ROI Schaefer atlas available in Nilearn, the probabilistic cerebellar atlas available in FSL, from [48], and the Oxford-GSK-Imanova connectivity striatal atlas from [49], also available in FSL. The cerebellar and striatal atlases were respectively composed of 28 and 7 ROIs, which was consistent with the 35 ROIs mentioned in the original paper. We merged the ROIs from the Schaefer, cerebellar and striatal atlas in this order to build a custom 135-ROI atlas which we used to extract regional features.

The three atlases were resampled to the whole-brain ReHo and fALFF maps using Nilearn and a nearest-neighbor interpolation, as done by the authors. Mean regional values for each imaging feature and parcellation were also extracted using Nilearn.

We obtained from the authors the custom atlas used in the original analyses. We found some slight differences between the cerebellar and striatal regions in the two atlases, e.g. in terms of size of the regions or division in subregions. We compared the mean regional values for the corresponding regions in the two atlases using paired two-sample t-tests. Among the 82 participants at baseline, 19 had significantly different values at $p < 0.05$ for fALFF and none at $p < 0.01$. Considering these small differences, we decided to report the results only using our atlas. Comparison of the two atlases is available in S1 Fig.

## Input features

**Clinical and demographic features.** In addition to imaging features, to better mirror clinical practices, the authors endeavored to integrated several clinical and demographic features as additional inputs to the machine-learning models. Clinical features included disease duration, symptom duration, dominant symptom side, Geriatric Depression Scale (GDS), Montreal Cognitive Assessment (MoCA), and presence of tremor, rigidity, or postural instability at Baseline. Baseline MDS-UPDRS score was also included as a feature when training models to predict outcomes at Year 1, Year 2, and Year 4. Demographic features included age, sex, ethnicity, race, handedness, and years of education.

We searched for the mentioned input features using the study files in the PPMI database, as done by the authors (see https://github.com/DeepLearningForPrecisionHealthLab/Parkinson-Severity-rsfMRI/blob/master/ppmiutils/dataset.py). For each feature, we searched for the corresponding columns in the study files and used the same character encoding method as the authors. The different features used and the methods to search and encode them for input to the models are shown in S1 Table.

To evaluate the robustness of the findings to different analytical conditions, we also compared the results obtained with different sets of features. In workflow *D.4—default workflow with only imaging features*, we trained models using only imaging features (regional measures of fALFF and ReHo), i.e., without clinical or demographic features. In workflow *D.3—default workflow with no imaging features*, we removed imaging features and trained models only on clinical and demographic features. Following an update of the PPMI database, the feature for dominant disease side was deprecated and only available as an archive file in the version of the database we had access to. We included the feature in the *default workflow* and removed it in another replication workflow, to assess the impact of this feature (*D.1—default workflow with no dominant disease side*). We did not contact the authors for the values of these features that they had downloaded, though they did factor prominently into their results, in order to better understand the relevance of the database update.

For models trained to predict MDS-UPDRS scores at Year 1, Year 2, and Year 4, Baseline MDS-UPDRS score was included as feature. However, due to the potential large effect of including this variable on the results, we trained a model with all features except this one and compared the performance of prediction models with and without the feature (*D.2—default workflow with no Baseline MDS-UPDRS*).

**Outcome measurement.** In [6], the authors used the above-mentioned imaging, clinical, and demographic features to predict MDS-UPDRS total scores. The MDS-UPDRS score consists of 4 parts with 51 items, each item values from 0 to 5. To compute the total scores, we summed the values of the 4 different parts available in PPMI study files. We used: MDS-UPDRS part Ia entered by a rater (PPMI column "NP1RTOT"), part Ib for the patient questionnaire (column "NP1PTOT"), part II ("NP2TOT"), part III ("NP3TOT") and part IV ("NP4TOT"). Missing values in "NP4TOT" columns were replaced with zeros, as done by the authors. There were no participants with missing values for the other parts of the score.

## Model selection and performance evaluation

We trained and optimized separate machine learning models to predict MDS-UPDRS scores from either ReHo or fALFF features, along with clinical and demographic features. Four machine learning models architectures were implemented using the latest version of scikit-learn at the time of this experiment, v1.3.0 [41], and were tested for each target-imaging feature (fALFF or ReHo) combination: ElasticNet regression, Support Vector Machine (SVM) with a linear kernel, Random Forest with a decision tree kernel, and Gradient Boosting with a

decision tree kernel. We recognize that this version of scikit-learn is likely newer than that used by the authors in 2022 and that we could download a prior version of scikit-learn, but did not because we wish to evaluate the relevancy of machine learning source code update. Each parcellation was also implemented, which resulted in 12 different combinations of model and parcellation per imaging feature and time point. All models were trained using our newer version of scikit-learn, we used the set of hyperparameters available in the authors code to train and optimize the models.

For hyperparameter optimization (1) and performance estimation (2), the authors used a nested cross-validation scheme, i.e., each model architecture × hyperparameter × parcellation combination was evaluated using (1) a 10-fold cross-validation inner-loop applied to the n-1 participants in the cohort and from which the combination with the lowest root mean squared error (RMSE) was selected, (2) a leave-one-out (LOO) cross-validation outer-loop where each iteration trained the selected model on all the participants in the cohort except one, and tested the model on the remaining held-out participant. To evaluate the impact of the evaluation pipeline on the results, we implemented a different nested cross-validation loop for model selection and evaluation for the *default workflow*. Fig 2 illustrates the different methods implemented. We evaluated the performance of each combination of model × parcellation separately: the 10-fold cross-validation inner-loop was used to select the set of hyperparameters (e.g. maximum tree depth for Random Forests) with the lowest RMSE, this set was used to train a model on all except one participants in the outer-loop and we tested the model on the held-out participant. Thus, we obtained performance estimates for each model × parcellation combination.

We also reported results obtained using the exact nested cross-validation scheme explained in the paper (*E.1—Workflow with paper's nested cross-validation*), i.e., the performance on each outer-fold is assessed with the best model × hyperparameter × parcellation combination

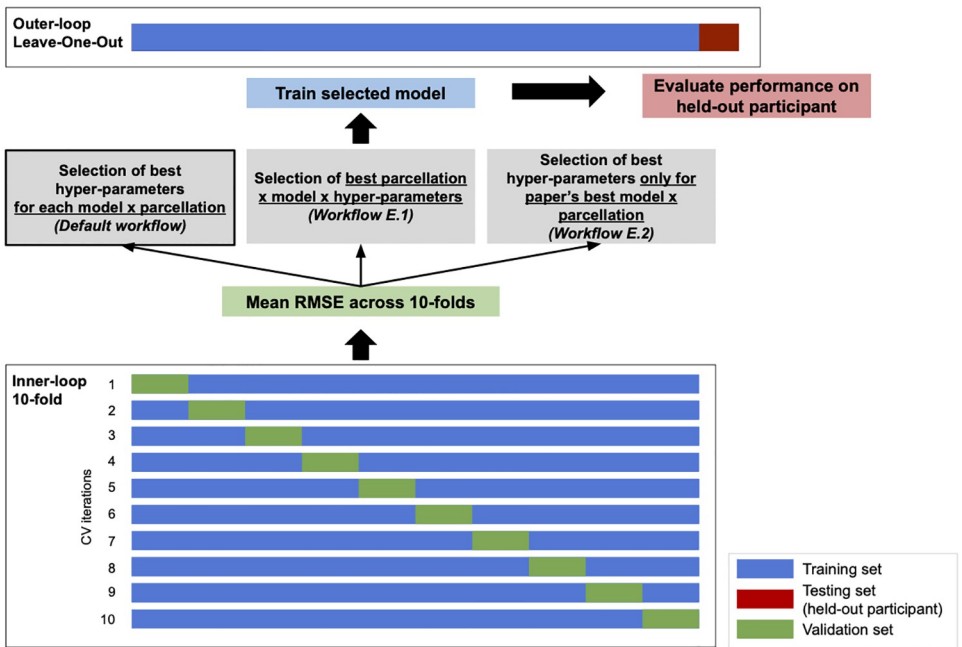

**Fig 2. Workflow of model selection and performance evaluation.** This workflow represents one iteration of the outer-loop with Leave-One-Out cross-validation and is iterated over all the dataset to estimate mean performance.

found on the 10-fold cross-validation of the inner-loop and averaged across outer-folds. Finally, as authors reported only the best performing model and parcellation for each imaging feature type and time point, we also reported the results we would have obtained had we only used the best model and parcellation reported in the paper (*E.2—Workflow with only paper's best model reporting*).

**Evaluation metrics.** As in the original paper, performance metrics included the coefficient of determination (R2), which represents the percentage of variance explained by the model, and the root mean squared error (RMSE), which represent the root mean squared difference between true and predicted values, as implemented in scikit-learn.

We defined a null performance to compare our R2 values to using permutation test. We fixed the model and parcellation scheme with ElasticNet and Schaefer atlas. This model and parcellation scheme were chosen as these were the most identified as best performing models across all time points and features in the original study [6]. We ran 1,000 permutations on the target labels and obtained performance for each feature and time point. At each permutation, we performed a nested cross-validation with 5-folds cross-validation as inner-loop and outer-loop. We optimized the hyper-parameter set of the model as done with the "real" models in the inner-loop and evaluated performance on the outer-loop. R2 values obtained using the different workflows were compared to this null performance to check if the models did not learn to predict only the average value.

We also compared the R2 values obtained with our different workflows with the original ones reported in [6]. We set a threshold of 0.15 to identify the workflows that were leading to important differences with the original ones. This threshold was chosen as it represents the lowest R2 reported across all experiments in the original study [6]. This means that this lowest reported R2 value was different from chance level with a threshold of 0.15. Thus, if we would have a difference higher than 0.15 compared to the original results, it would also mean that we would possibly obtain chance results. Moreover, this threshold was considered sufficiently high for the original authors to say that the model was making good predictions. Thus, we kept this threshold to compare the performance of our models with the original ones.

To evaluate the models' ability to classify high versus low severity participants, as it was performed in the original study [6], a threshold was set to separate the participants and each model's predictions were thresholded post-hoc. This threshold was computed by using the average of the median MDS-UPDRS score at each of the four time points. In [6], the threshold was 35. We computed this threshold the same way for the replication cohort and for the closest-to-original cohort. We obtained a value of 36 for the replication cohort and 35 for the closest-to-original one. Authors also mentioned having found no significant difference (p >0.05) between the high and low-severity groups in motor predominance (Part III score as a percentage of total score) at each time point. With our thresholds, we ran two sample t-tests between high and low severity groups in the two cohort and did not find any significant difference with $\alpha$ = 0.05 either in any cohort or time point. Performance metrics for this secondary classification outcome included area under the receiver operating characteristic curve (AUC), positive predictive value (PPV), negative predictive value (NPV), specificity, and sensitivity.

**Authors derivatives.** At the second step of the study (Phase 2), authors shared with us the derived data used in the original study (i.e. whole-brain fALFF and ReHo maps for the original cohort). We applied the input features selection (clinical and demographics) and machine learning model training and selection to these data and computed the results for the *Workflow 0*. While we could also have asked the authors for their original image processing pipeline, the retrieval of the exact version of the pipeline and software packages is challenging. The direct use of derived data allowed us to verify the reproduction of these steps and to get more information on the potential factors of variations in the results (e.g., suppressing differences in

cohort selection and imaging processing, while retaining some potential differences in the version of scikit-learn). This reproduction attempt will be referred to as "partial reproduction": we used the derived data provided by the authors and thus, did not attempt to reproduce the cohort selection and pre-processing steps. In that case, we consider that the data are the same as the ones used in the original study. Thus, any divergence between our results and the original ones would be explained by potential differences in the version of scikit-learn, the non-determinism of algorithms, or differences in computer environments.

### Feature importance

As in [6], we measured feature importance in the models trained for each time point and imaging feature (fALFF or ReHo). For the ElasticNet and SVM models, we used the coefficients of the trained models to determine feature importance, since coefficients of higher magnitude indicate more important features in these two models. The sign of the coefficient was indicative of whether the feature was positively or negatively associated with the prediction target. For Random Forest and Gradient Boosting models, we used impurity-based feature importance coupled with univariate linear correlation to determine the direction of the association. Feature importance was computed on each iteration of the outer-loop and the median importance was reported for each feature.

To name the imaging features, we used the same method as the authors of [6]: the centroid of each feature's ROI was computed, if the feature was located in a ROI of the Automated Anatomical Labeling (AAL) atlas [50], this label was allocated to the ROI. If not, we searched for the nearest ROI of the AAL atlas. Authors also sent us their ROI labels. However, since we decided to use the reproduced Schaefer atlas, we used the reproduced labels in the figures for consistency.

## Results

### Cohort selection

Using the method described above, we built two cohorts from the PPMI database: the replication cohort and the closest-to-original cohort.

Table 2 shows the demographics and Baseline clinical characteristics of the replication and closest-to-original cohorts compared to the original cohort reported in [6]. The replication cohort was composed of respectively 102, 67, 61 and 46 participants for time points Baseline, Year 1, Year 2, and Year 4. The closest-to-original cohorts at the same time points were composed of respectively 82, 51, 41 and 30 participants. To evaluate the differences between the original cohort and our two cohort, we determined whether the values obtained for our cohorts fell within a ±10% range of the original value. Values that do not fulfill these criteria are in Bold text in Table 2.

Compared to the original cohort, our replication cohort showed similar demographics characteristics at each time point, except at Year 4 where our replication cohort showed a higher age on average than in the original cohort (66.2 ± 10.1 years compared to 59.5 ± 11.0). Regarding clinical variables, mean MoCA score, GDS total score and Hoehn-Yahr stage were similar between the two cohorts at all time points. However, we found higher mean disease durations in the replication cohort than in the original one at all time points, for instance at Baseline with (866.9 days ± 598.7 days) in replication vs (770 days ± 565 days) in original. We also observed lower baseline mean MDS-UPDRS scores in the replication cohort for all time points except Baseline, in particular for Year 2 with a mean baseline score of 35.2 ± 16.1 compared to 40.2 ± 18.2 in the original cohort. For the two time points Year 2 and Year 4 where we mostly found differences, even if mean Baseline scores in the replication cohort differed from

**Table 2. Demographic and clinical variables for the different cohorts.** Orig. = original paper cohort. Repli. = replication cohort. Closest = closest-to-original cohort. Values are reported in percentages of the cohort or in mean values ± standard deviation. **Bold text** refers to features for which a meaningful difference was observed compared to the original cohort.

| | Baseline | | | Year 1 | | | Year 2 | | | Year 4 | | |
|---|---|---|---|---|---|---|---|---|---|---|---|---|
| | Orig. | Repro. | Closest | Orig. | Repro. | Closest | Orig. | Repro. | Closest | Orig. | Repro. | Closest |
| % Caucasian | 95.1 | 95.1 | 93.9 | 94.4 | 94.0 | 94.1 | 97.8 | 95.1 | 95.1 | 97.0 | 97.8 | 96.7 |
| % African-American | 2.4 | 2.0 | 2.4 | 1.9 | 1.5 | 0.0 | 0 | 1.6 | 0.0 | 0 | 0.0 | 0.0 |
| % Asian | 3.7 | 2.9 | 3.7 | 5.6 | 4.5 | 5.9 | 4.4 | 3.3 | 4.9 | 3.0 | 2.2 | 3.3 |
| % Hispanic | 1.2 | 1.0 | 0.0 | 0 | 1.5 | 0.0 | 0 | 1.6 | 0.0 | 0 | 0.0 | 0.0 |
| % Male | 67.0 | 66.7 | 67.1 | 68.5 | 65.7 | 68.6 | 82.2 | 80.3 | 85.4 | 75.8 | 67.4 | 73.3 |
| % right-handed | 89.0 | 89.2 | 89.0 | 85.2 | 85.1 | 84.3 | 88.9 | 90.2 | 90.2 | 87.9 | 84.8 | 86.7 |
| Mean age, years | 62.1 ± 9.8 | 62.0 ± 9.5 | 62.1 ± 9.7 | 61.9 ± 10.3 | 62.2 ± 9.9 | 63.0 ± 10.4 | 63.6 ± 9.2 | 64.7 ± 9.1 | 65.9 ± 9.4 | 59.5 ± 11.0 | **66.2 ± 10.1** | **63.8 ± 11.0** |
| Mean years of education | 15.6 ± 3.0 | 15.6 ± 2.8 | 15.7 ± 2.9 | 15.1 ± 3.2 | 15.5 ± 2.9 | 15.4 ± 2.9 | 15.1 ± 3.3 | 15.4 ± 2.8 | 15.5 ± 3.0 | 15.0 ± 3.4 | 15.3 ± 3.0 | 15.2 ± 3.4 |
| Mean disease duration at Baseline, days | 770 ± 565 | **866.9 ± 598.7** | 760.3 ± 559.2 | 808 ± 576 | **904.1 ± 614.5** | 808.5 ± 580.0 | 771 ± 506 | **867.5 ± 516.3** | 732.0 ± 462.8 | 532 ± 346 | **746.6 ± 624.6** | **464.6 ± 294.9** |
| Mean MDS-UPDRS at Baseline | 33.9 ± 15.8 | 34.5 ± 15.6 | 33.9 ± 16.1 | 38.0 ± 20.9 | **33.4 ± 15.1** | 34.1 ± 15.4 | 40.2 ± 18.2 | **35.0 ± 15.1** | **35.2 ± 16.1** | 34.9 ± 15.7 | **30.7 ± 13.9** | **26.1 ± 11.4** |
| Mean MDS-UPDRS at timepoint | - | - | - | 39.2 ± 21.6 | 40.7 ± 24.5 | 39.9 ± 22.0 | 40.9 ± 18.5 | 40.0 ± 18.7 | 40.7 ± 18.7 | 35.9 ± 16.5 | **41.5 ± 19.8** | 34.2 ± 16.2 |
| Mean MoCA at Baseline | 26.7 ± 2.8 | 26.5 ± 3.0 | 26.4 ± 2.8 | 26.9 ± 3.2 | 27.0 ± 2.9 | 26.7 ± 3.1 | 26.7 ± 3.5 | 27.0 ± 2.5 | 26.5 ± 2.4 | 27.5 ± 2.3 | 26.8 ± 3.2 | 27.4 ± 2.6 |
| Mean GDS at Baseline | 5.4 ± 1.4 | 5.4 ± 1.4 | 5.4 ± 1.5 | 5.4 ± 1.6 | 5.5 ± 1.8 | 5.5 ± 1.9 | 5.4 ± 1.2 | 5.5 ± 1.3 | 5.6 ± 1.3 | 5.4 ± 1.7 | 5.8 ± 1.8 | 5.6 ± 1.7 |
| Mean Hoehn-Yahr stage | 1.8 ± 0.5 | 1.7 ± 0.5 | 1.7 ± 0.5 | 1.8 ± 0.5 | 1.8 ± 0.6 | 1.7 ± 0.5 | 1.8 ± 0.5 | 1.9 ± 0.5 | 1.9 ± 0.5 | 1.7 ± 0.5 | 1.9 ± 0.5* | 1.8 ± 0.5 |
| Number of subject | 82 | 102 | 82 | 53 | 67 | 51 | 45 | 61 | 41 | 33 | 46 | 30 |

the original ones, mean MDS-UPDRS scores at prediction time point were more similar to the original one. At Year 4, however, we also found a higher mean MDS-UPDRS score at prediction time point (30.7 ± 13.9) than in the original cohort.

The closest-to-original cohort, obtained using the same cohort selection method as in the original study, exhibited almost the same characteristics as the original one at Baseline. For subsequent time points, we found some differences, in particular at Year 2 and at Year 4: participants were older in the closest-to-original cohort than in the original study at Year 4 (63.8 ± 11.0 in the closest to original cohort compared to 62.1 ± 9.8 in the original), Baseline mean MDS-UPDRS score was lower for Year 2 (40.2 ± 18.2 in original, 35.2 ± 16.1 in closest-to-original) and Year 4 (34.9 ± 15.7 in original, 26.1 ± 11.4 in closest-to-original) and mean MDS-UPDRS score at prediction time point was similar to the original cohort except at Year 4.

For time points Year 1, Year 2, and Year 4, we were not able to find all the participants that were included in the original cohort: the patients included in our closest-to-original cohorts represented respectively 96% (Year 1), 91% (Year 2) and 91% (Year 4) of the patients included in the original cohort. However, only represented 76% (Year 1), 67% (Year 2), and 65% (Year 4) of the replication cohort was composed of patients of the original cohort.

Fig 3 compares the distribution of MDS-UPDRS scores in our two cohorts with the one in the original cohort reported in S1 Fig in [6]. Distributions of MDS-UPDRS scores at Baseline were similar between our two cohorts but seemed different from the original cohort one. The observed difference between the original and closest-to-original distributions might result

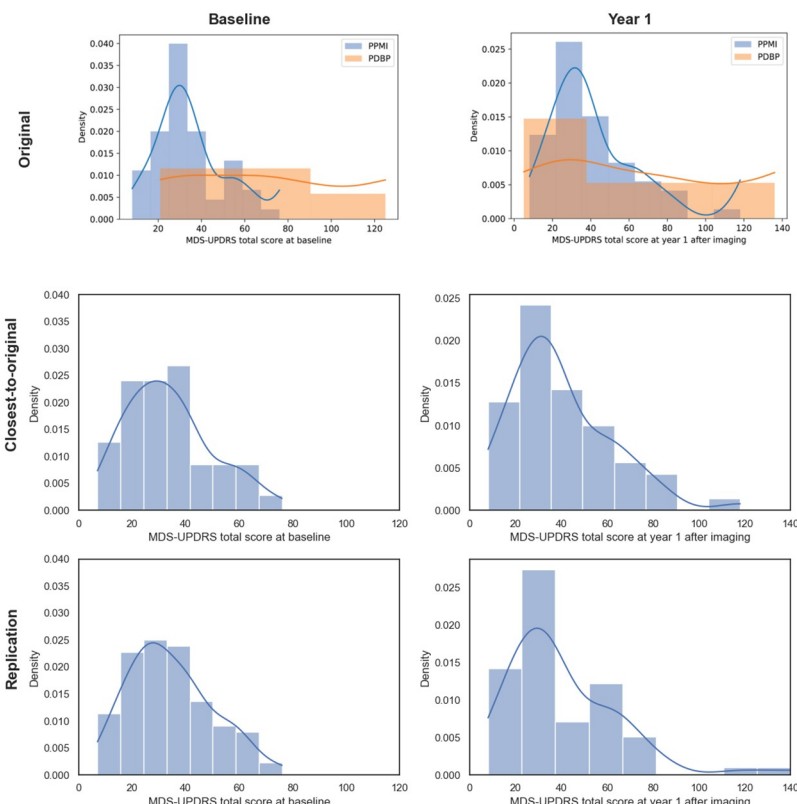

**Fig 3. Distribution of MDS-UPDRS scores reported in the original paper's cohort (*top*: S1 Fig extracted from [6]), the replication cohort (*middle*) and the closest-to-original cohort (*bottom*).**

from the fact that different sessions were used for 4 of the participants in the closest-to-original cohort compared to the original one. At Year 1, however, the closest-to-original cohort presented an MDS-UPDRS score distribution more similar to the original one than the replication one, suggesting that the differences at Baseline did not originate in differences in MDS-UPDRS score calculations. We found no significant difference between the distribution of MDS-UPDRS scores in the replication and closest-to-original cohort neither at Baseline nor at Year 1 using Kolmogorov-Smirnov distribution testing.

## Image quality control

After running all the pre-processing pipelines, we checked the resulting images and looked for potential pipeline failures. Regarding registration, all participants brains were correctly registered to the MNI space after visual inspection. Brain masking was also successful for most of the participants, except for 2 in which we found a small artifact in the inter-hemispheric area. Given the low magnitude of this artefact and its location, we decided to keep these two participants in the study.

Most participants of the study showed high movement parameters. Indeed, out of 102, 80 showed at least one time point with a frame-wise displacement superior to 0.5mm. The mean frame-wise displacements across all time points for each participant are reported in S1 and S2 Tables. The mean frame-wise displacement across time points and participants was of 0.258. However, since the authors in [6] did not remove high-motion volumes within participants, that removing volumes entirely can disrupt some derived values, and that completely removing participants with high-motion volumes would highly decrease our cohort's sample size, we chose to keep all participants and all volumes.

Regarding segmentation masks, after visual inspection no significant artifact was found for any participants using AFNI segmentation in default workflow. For some participants, small distortions were found in particular close to brain extremities (inter-hemispheric area or close to the skull in occipital and parietal regions). Using FSL segmentation however, we found brain masking issues that had impacts on segmentation quality. We used BET using default parameters to skullstrip images before segmentation and since we chose to explore the impact of different default implementations of pipelines, we did not exclude the segmentations for any participant nor segmentation workflow.

With the fMRIprep pipeline, observations were similar regarding movement parameters and registration. There was no large artefact in the segmentation masks.

## Performance of the *default workflow*

The first objective of this study was to re-implement the models described in [6] and to compare their performance with the one in the original study. In the default workflow, we implemented the default choices described in Fig 1: closest-to-original cohort, image pre-processing pipeline with AFNI segmentation, z-scoring of whole-brain fALFF and ReHo maps, use of all demographic, clinical and imaging features described in the original paper, and the model selection method derived from the authors' code.

We trained 12 models per time point (Baseline, Year 1, Year 2, Year 4) and imaging feature (fALFF or ReHo), corresponding to 4 machine learning models × 3 brain parcellations. We reported for each imaging feature and time point the performance of the 12 models in Table 3.

Chance levels were computed using permutation tests as described in the Evaluation metrics section. We obtained R2 values that represented the chance prediction performance at different time point for fALFF and ReHo. These values are also presented in Table 3.

**Table 3. Predictive performance achieved for each MDS-UPDRS time point and each imaging feature type, computed through leave-one-out cross-validation using the default workflow ("Repli.").** Metric: R2, coefficient of determination. Green cells corresponds to original performance reported in [6]; Cyan cells corresponds to best performance achieved using the default workflow; Red cells corresponds to chance level computed using permutation test.

| Time | Feature | Type | ElasticNet | | | SVM | | | GradientBoosting | | | RandomForest | | |
|---|---|---|---|---|---|---|---|---|---|---|---|---|---|---|
| | | | schaefer | basc197 | basc444 | schaefer | basc197 | basc444 | schaefer | basc197 | basc444 | schaefer | basc197 | basc444 |
| Baseline | fALFF | Orig. | | | | | | | 0.242 | | | | | |
| | | Repli. | 0.04 | -0.035 | -0.045 | -0.718 | -0.241 | -0.182 | -0.039 | 0.205 | 0.061 | -0.024 | 0.068 | 0.02 |
| | | Null | -0.041 | | | | | | | | | | | |
| | ReHo | Orig. | | | | | | | 0.304 | | | | | |
| | | Repli. | 0.124 | 0.057 | 0.117 | -0.3 | -0.4 | -0.152 | -0.102 | 0.028 | 0.027 | 0.024 | 0.022 | 0.099 |
| | | Null | -0.036 | | | | | | | | | | | |
| Year 1 | fALFF | Orig. | 0.558 | | | | | | | | | | | |
| | | Repli. | 0.453 | 0.717 | 0.5 | 0.519 | 0.216 | 0.185 | 0.622 | 0.575 | 0.506 | 0.369 | 0.499 | 0.444 |
| | | Null | -0.079 | | | | | | | | | | | |
| | ReHo | Orig. | 0.453 | | | | | | | | | | | |
| | | Repli. | 0.535 | 0.434 | 0.512 | 0.04 | -0.094 | -0.01 | 0.36 | 0.261 | 0.289 | 0.442 | 0.392 | 0.393 |
| | | Null | -0.077 | | | | | | | | | | | |
| Year 2 | fALFF | Orig. | 0.463 | | | | | | | | | | | |
| | | Repli. | 0.529 | 0.277 | 0.285 | -0.031 | 0.108 | -0.413 | -0.19 | 0.08 | 0.01 | 0.138 | 0.206 | 0.09 |
| | | Null | -0.101 | | | | | | | | | | | |
| | ReHo | Orig. | 0.471 | | | | | | | | | | | |
| | | Repli. | 0.344 | 0.191 | 0.287 | -0.915 | -0.741 | -0.051 | -0.03 | 0.001 | -0.033 | 0.267 | 0.121 | 0.251 |
| | | Null | -0.094 | | | | | | | | | | | |
| Year 4 | fALFF | Orig. | | | | | 0.152 | | | | | | | |
| | | Repli. | 0.397 | 0.115 | 0.351 | 0.196 | -0.134 | -0.296 | 0.08 | 0.411 | -0.355 | 0.079 | 0.338 | 0.01 |
| | | Null | -0.129 | | | | | | | | | | | |
| | ReHo | Orig. | | | | | 0.255 | | | | | | | |
| | | Repli. | 0.072 | 0.09 | -0.175 | -0.12 | -0.23 | -0.139 | -0.017 | 0.312 | 0.041 | -0.007 | 0.02 | 0.0 |
| | | Null | -0.141 | | | | | | | | | | | |

Using the default workflow, we obtained prediction scores different but relatively consistent with the results of [6], for all models × parcellation combination. At Baseline, our best model performed better than chance and we obtained a R2 value close to the one reported in the original paper with the best model. However, the best-performing models were different from those reported in the original study: instead of Schaefer atlas and Gradient Boosting for both fALFF and ReHo features, we found for fALFF the Gradient Boosting Regressor with BASC197 atlas, with R2 = 0.205 (original R2 = 0.242) and ElasticNet and Schaefer for ReHo with R2 = 0.124 (original R2 = 0.304).

At Year 1, the performance of our models was better than reported in the original study, with an increase of the R2 of 0.16 and 0.08 for fALFF and ReHo respectively. For other time points (Year 2 and Year 4), results were slightly different from those reported in [6] but overall consistent. These differences were not constant between ReHo and fALFF at Year 2, but were similar at Year 4: for fALFF, we obtained higher R2 scores than in the original study at Year 2 and at Year 4 (0.529 and 0.397 compared to 0.463 and 0.152 in the original paper); for ReHo, we obtained lower R2 scores than in the original ones at Year 2 (0.344 instead of 0.471) and higher R2 scores at Year 4 (0.312 compared to 0.255 in the original study). For these two time points, the mean MDS-UPDRS scores at Baseline were significantly different between the original cohort and our closest-to-original cohort, which might explain these differences in

performance. In this context, the results observed remained similar in terms of effect size and replication remained satisfactory.

At each time point, the best model x parcellation combination performed better than chance-level. Some of the combinations led to very low performance, for instance SVM with Schaefer atlas at Year 2. At every time point and with every feature (except at Year 1 with fALFF), at least one combination gave a performance lower than chance.

## Authors derivatives

In Fig 4, we can see that using authors derivatives and thus, the original cohort, we achieve performance that are very close to the original ones, except at Year 4 for which performances are higher. This informs us on the quality of the reproduction of the clinical and demographic features selection, but also on the machine learning models training and selection.

## Robustness to workflow variations

We assessed the performance of the different models for each time point and feature for different variations of the default workflow (which itself, corresponds to a replication of the original workflow) (Fig 4).

*Workflow A.2*, in which we trained the different models on the replication cohort instead of the closest-to-original one, showed only small differences in R2 values with the *default work-flow*, except for fALFF at Year 1 and ReHo at Year 4. Indeed, performance was slightly lower at Year 1 for fALFF and higher at Year 4 for ReHo, with raw effect size above 0.15. At Year 1, the replication cohort was composed of 16 more participants than the closest-to-original cohort and exhibited a lower mean MDS-UPDRS score at Baseline compared to the original cohort. At Year 4, we also found differences in term of sample size, age of participants and Baseline MDS-UPDRS score between the replication cohort, the original one and the closest-to-original one. These differences might explain the variations between performance of models, even if R2 values remained better-than-chance for Year 1 and close to other performance obtained with different variations. Best performance of *workflow A.2* remained better than chance-level.

Performance of models trained with variations in pre-processing pipeline (*workflows B.1, B.2 and B.3*) was similar to those of the default workflow, with R2 absolute difference with the *default workflow* below 0.15 except at Year 4 with fALFF in which the *B.2 workflow* (no structural segmentation) led to lower R2 values and at baseline with fMRIprep pipeline (*B.3 work-flow*). For these, the best performance achieved was better than chance.

Regarding the impact of feature computation variations (*workflows C.1 and C.2*), we found better performance at Baseline for workflows *C.2—default workflow with ALFF* in which the best model × parcellation combination led to a better R2 value than the one reported in the original study (0.325 vs 0.242 in the original paper). We also observed this phenomenon with the *C.1 workflow* in which we used non z-scored ReHo maps: we found a higher performance than the one obtained with the default workflow and reported in the original study ($R2 = 0.374$). For these two variations, R2 differences with default remained lower than 0.1. At Year 1 and Year 4 with fALFF however, the use of ALFF instead of fALFF (*workflow C.2*) led to lower performance (R2 mean absolute difference above 0.15). This observation was not found at Year 2.

For Year 1 and Year 2 predictions, the set of input features (*workflows D.*) had a large impact on the performance of these models. In particular, models trained without Baseline MDS-UPDRS score (D.2) and with only imaging features (D.4) showed lower R2 values for fALFF and for ReHo at Year 1 and Year 2 (R2 absolute difference above 0.2), which suggests that Baseline MDS-UPDRS played a central role in the prediction of MDS-UPDRS at follow-up visits compared to imaging features. It also explains why variations in the extraction of

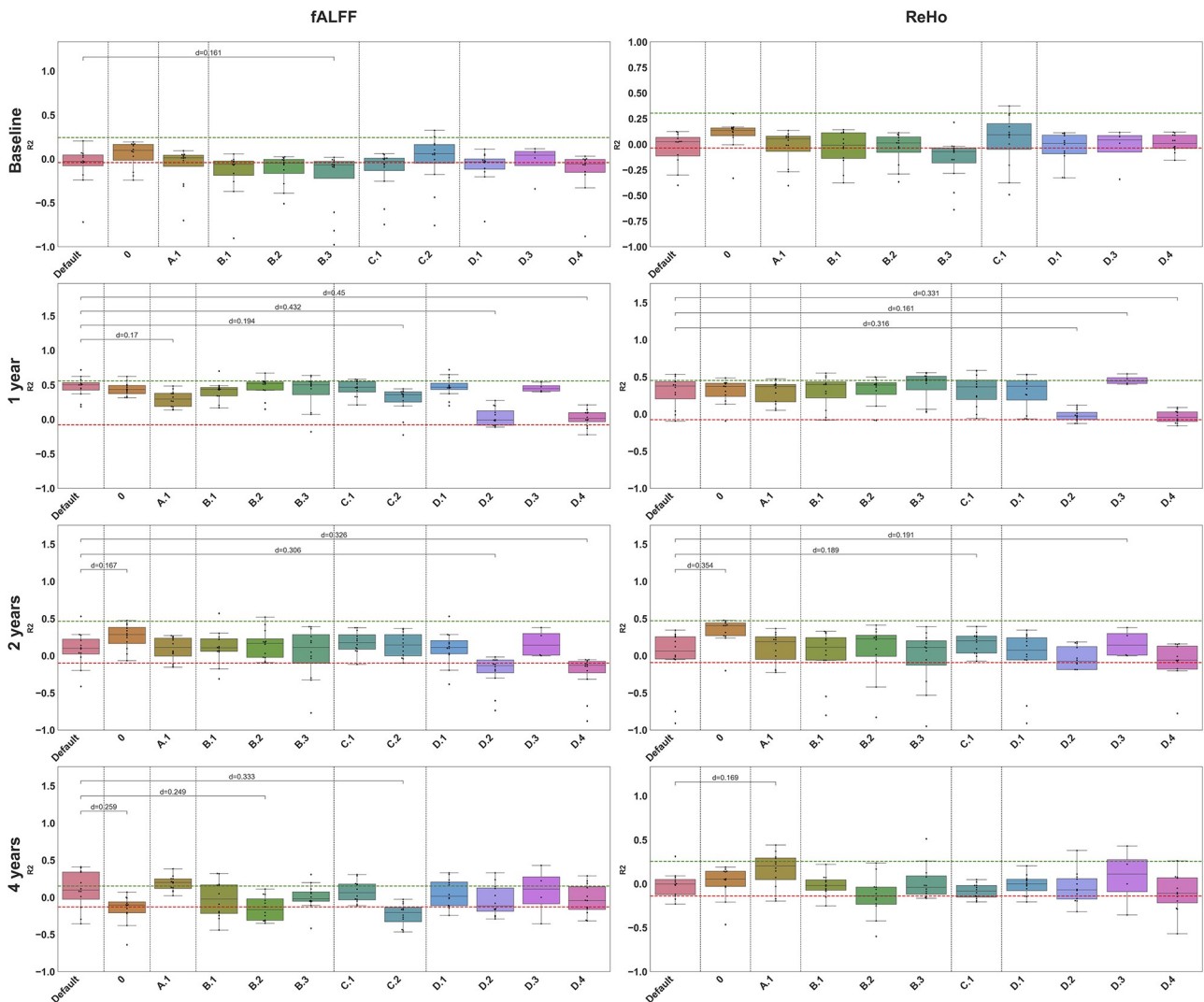

**Fig 4. Performance of models trained for prediction at each time point, using fALFF or ReHo, with variations in the workflow.** Boxes represent the performance (R2 values) of the 12 models (4 models × 3 parcellations). Green horizontal dashed lines show the R2 value reported in the original study for the corresponding time point and feature. Red horizontal dashed lines show the chance-level computed using permutation test. Raw effect sizes (d) are computed as absolute difference between the mean R2 performance with *default workflow* and mean R2 performance with other variations. Only large differences (above threshold *d* = 0.15) are reported.

- *Workflow 0*—reproduction using authors derivatives.

- *Workflow A.1*—variation of the default workflow with replication cohort.

- *Workflow B.1*—variation of the default workflow with FSL segmentation,

- *Workflow B.2*—variation of the default workflow without structural priors,

- *Workflow B.3*—variation of the fMRIprep pipeline.

- *Workflow C.1*—variation of the default workflow with no Z-scoring,

- *Workflow C.2*—variation of the default workflow with ALFF.

- *Workflow D.1*—variation of the default workflow with no dominant disease side,

- *Workflow D.2*—variation of the default workflow with no Baseline MDS-UPDRS,

- *Workflow D.3*—variation of the default workflow with no imaging features,

- *Workflow D.4*—variation of the default workflow with only imaging features.

- *Workflow E.1*—variation of the default workflow with paper's nested cross-validation,

- **Workflow E.2**—variation of the default workflow with only paper's best model reporting.

imaging features (pre-processing or computation) only had a lower impact on the performance for these two time points.

Overall, at Year 1 and Year 2, performance seemed to be driven mostly by clinical and demographic features, in particular by MDS-UPDRS Baseline scores. At Baseline and Year 4, other variations related to image features (pre-processing and feature computation) were associated with larger changes in performance. For all workflows, time points and feature, best performing model x parcellation combination always exhibited better than chance performance.

## Model choice and performance reporting

Table 4 compares the results obtained using different model selection and evaluation methods. Using the nested cross-validation described in the paper (*Workflow E.1*), we obtained lower results than the original ones and than the ones obtained with our best models for all time points (for instance, $R2 = 0.049 vs 0.205$ with our best model for prediction with fALFF at Baseline). Using this method, the models at Year 1 and Year 2 were still well performing compared to other time point, for both ReHo and fALFF, with particularly high R2 values (between around 0.4 and 0.6) obtained using any reporting method.

Results computed using the same model and parcellation as the best performing combinations in the original paper (Table 2 from [6]) (*Workflow E.2*) also had lower performance than in the original study, for all time points (e.g. $R = −0.102$ for prediction with ReHo at Baseline). However, as observed for nested cross-validation, the performance obtained with these models at Year 1 and Year 2 was still high and close to the ones obtained with our best models. We speculate that the effect size detected with models at these time points was large and thus, tended to be more reproducible across optimization schemes.

In [6], authors also report the model's ability to classify high- versus low-future severity subjects. The performance obtained for this task was consistent with the observation made on R2 values: models with high performance in terms of R2 were usually good at distinguishing high and low severity patients (e.g., AUC of 0.805 and 0.767 for prediction at Year 1 with respectively fALFF and ReHo using the *default workflow*).

## Feature importance

To further explore the reproducibility and replicability of findings in [6], we measured feature importance for the ReHo and fALFF imaging features and the default workflow, across all time points. Figs 5 and 6 compare the feature importances obtained with the *default workflow* to the ones reported in the original study.

Feature importance showed relatively few overlap between the ones obtained using our default workflow and those reported in the original study, especially for imaging features, at all time points. Note that the same mask Schaefer atlas that was used by [6] was not used here. For instance, for fALFF at Baseline, the left postcentral region was identified as the most important feature for prediction in our study and was not identified in the original study. For ReHo, we found no important imaging feature that was similar to the ones detected in the

**Table 4. Performance reported using different model selection and evaluation methods.** "Original" is the performance reported in the Original study [6]. "Default" is the performance obtained with the model × parcellation that obtained the best performance with our default workflow. "Workflow E.1" is the performance obtained when using the nested cross-validation scheme described in the paper (i.e. optimizing model × parcellation in the inner fold). "Workflow E.2" is the performance obtained with the model and parcellation reported in the paper.

| Time point | Feature | Type | R2 | RMSE | AUC | PPV | NPV | Spec. | Sens. |
|---|---|---|---|---|---|---|---|---|---|
| Baseline | fALFF | Original | 0.242 | 14.006 | 0.668 | 60.0% | 74.0% | 75.5% | 58.1% |
| | | Default | 0.205 | 14.26 | 0.584 | 51.7% | 66.0% | 71.4% | 45.5% |
| | | Workflow E.1 | 0.049 | 15.6 | 0.514 | 42.3% | 60.7% | 69.4% | 33.3% |
| | | Workflow E.2 | -0.039 | 16.31 | 0.493 | 39.4% | 59.2% | 59.2% | 39.4% |
| | ReHo | Original | 0.304 | 13.415 | 0.674 | 59.4% | 75.0% | 73.5% | 61.3% |
| | | Default | 0.124 | 14.98 | 0.716 | 63.9% | 78.3% | 73.5% | 69.7% |
| | | Workflow E.1 | -0.164 | 17.26 | 0.528 | 43.8% | 62.0% | 63.3% | 42.4% |
| | | Workflow E.2 | -0.102 | 16.8 | 0.493 | 39.3% | 59.3% | 65.3% | 33.3% |
| Year 1 | fALFF | Original | 0.558 | 14.256 | 0.753 | 70.4% | 80.0% | 71.4% | 79.2% |
| | | Default | 0.717 | 11.6 | 0.805 | 75.9% | 86.4% | 73.1% | 88.0% |
| | | Workflow E.1 | 0.569 | 14.3 | 0.786 | 73.3% | 85.7% | 69.2% | 88.0% |
| | | Workflow E.2 | 0.453 | 16.11 | 0.69 | 62.9% | 81.2% | 50.0% | 88.0% |
| | ReHo | Original | 0.453 | 15.861 | 0.753 | 70.4% | 80.0% | 71.4% | 79.2% |
| | | Default | 0.535 | 14.85 | 0.767 | 71.0% | 85.0% | 65.4% | 88.0% |
| | | Workflow E.1 | 0.483 | 15.67 | 0.726 | 70.4% | 75.0% | 69.2% | 76.0% |
| | | Workflow E.2 | 0.535 | 14.85 | 0.767 | 71.0% | 85.0% | 65.4% | 88.0% |
| Year 2 | fALFF | Original | 0.463 | 13.426 | 0.765 | 78.6% | 76.5% | 68.4% | 84.6% |
| | | Default | 0.529 | 12.68 | 0.669 | 69.2% | 66.7% | 55.6% | 78.3% |
| | | Workflow E.1 | 0.478 | 13.35 | 0.669 | 69.2% | 66.7% | 55.6% | 78.3% |
| | | Workflow E.2 | 0.529 | 12.68 | 0.669 | 69.2% | 66.7% | 55.6% | 78.3% |
| | ReHo | Original | 0.471 | 13.322 | 0.739 | 75.9% | 75.0% | 63.2% | 84.6% |
| | | Default | 0.344 | 14.95 | 0.635 | 65.5% | 66.7% | 44.4% | 82.6% |
| | | Workflow E.1 | 0.272 | 15.76 | 0.607 | 63.3% | 63.6% | 38.9% | 82.6% |
| | | Workflow E.2 | 0.344 | 14.95 | 0.635 | 65.5% | 66.7% | 44.4% | 82.6% |
| Year 4 | fALFF | Original | 0.152 | 14.957 | 0.636 | 64.7% | 62.5% | 62.5% | 64.7% |
| | | Default | 0.411 | 12.19 | 0.833 | 91.7% | 77.8% | 93.3% | 73.3% |
| | | Workflow E.1 | 0.242 | 13.83 | 0.733 | 73.3% | 73.3% | 73.3% | 73.3% |
| | | Workflow E.2 | -0.134 | 16.92 | 0.633 | 66.7% | 61.1% | 73.3% | 53.3% |
| | ReHo | Original | 0.255 | 14.015 | 0.699 | 73.3% | 66.7% | 75.0% | 64.7% |
| | | Default | 0.312 | 13.18 | 0.667 | 72.7% | 63.2% | 80.0% | 53.3% |
| | | Workflow E.1 | -0.044 | 16.23 | 0.567 | 60.0% | 55.0% | 73.3% | 40.0% |
| | | Workflow E.2 | -0.23 | 17.62 | 0.6 | 63.6% | 57.9% | 73.3% | 46.7% |

original study. However, for some brain regions for which an imaging feature was identified as an important feature, hemispheric opposites or sub-parts of the same global regions were identified in our models compared to the original detected features. For instance, the middle cingulum was identified in our Baseline model with ReHo but in the left hemisphere instead of the right one in the original paper. For this model, regions of the frontal cortex were also detected as important in the original paper, but those we found were very close or were part of the same lobe/region (e.g. frontal supero-orbital and middle in original, frontal inferior in ours). Regions identified for fALFF and ReHo were also different at Baseline, consistently with the findings of [6].

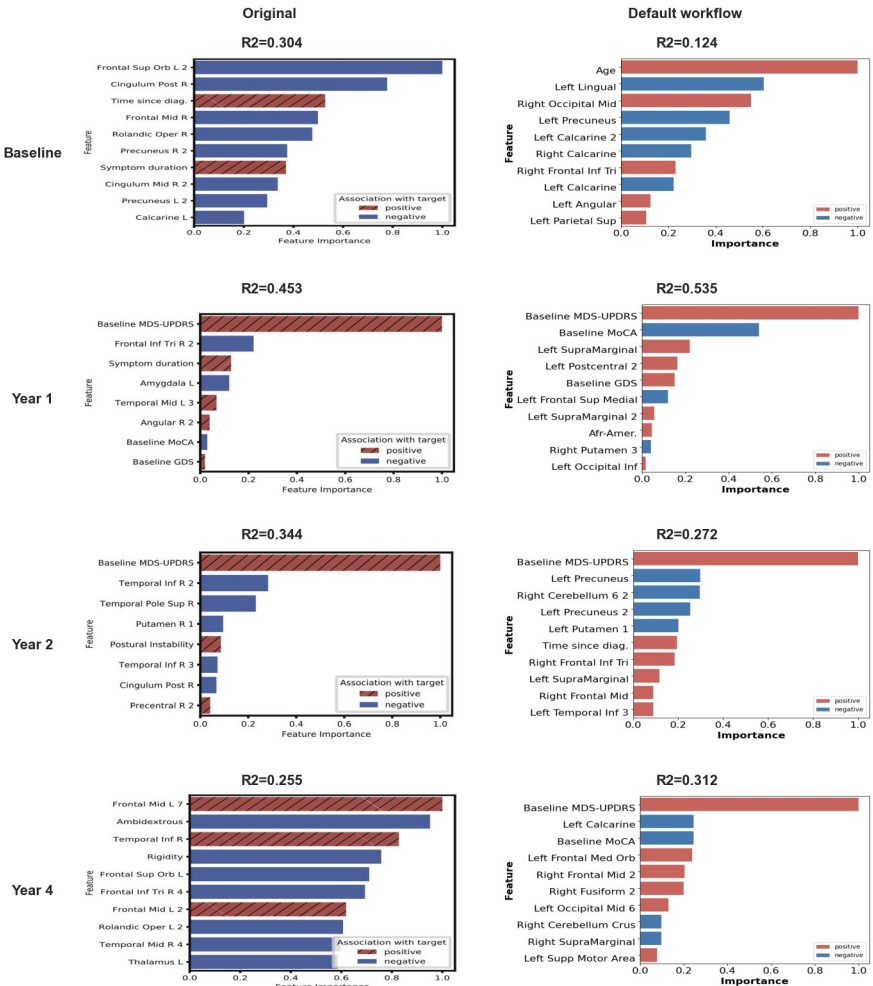

**Fig 5. Predictive features learned by the best performing models to predict MDS-UPDRS score at each time point for the original study (left—extracted from [6]) and the *default workflow* (right) using ReHo.** Features with low importance were not shown. Red bars indicate a positive association and Cyan bars indicate a negative association. Stars (*) represent the presence of this feature in the original study and the default workflow.

For other time points, the main feature of importance was the Baseline MDS-UPDRS score for both fALFF and ReHo and other features had a lower importance value, in particular at Year 1 and at Year 2. This observation was also supported by the performance of models that did not include the Baseline MDS-UDPRS score in their feature set: these models showed lower performance at these two time points compared to the default models ($p < 0.01$). Note that, as shown in Figs 5 and 6, similar R2 is attained, though through different sets of features.

## Discussion

### Summary

We investigated the reproducibility and replicability of the predictive models of PD progression described in [6]. Using the *default workflow*, i.e., with a cohort closest to the one described in [6] and a workflow with the fewer possible variations from the original one, the performance of our best models was better than chance ($R2 > 0$). For both ReHo and fALFF, we

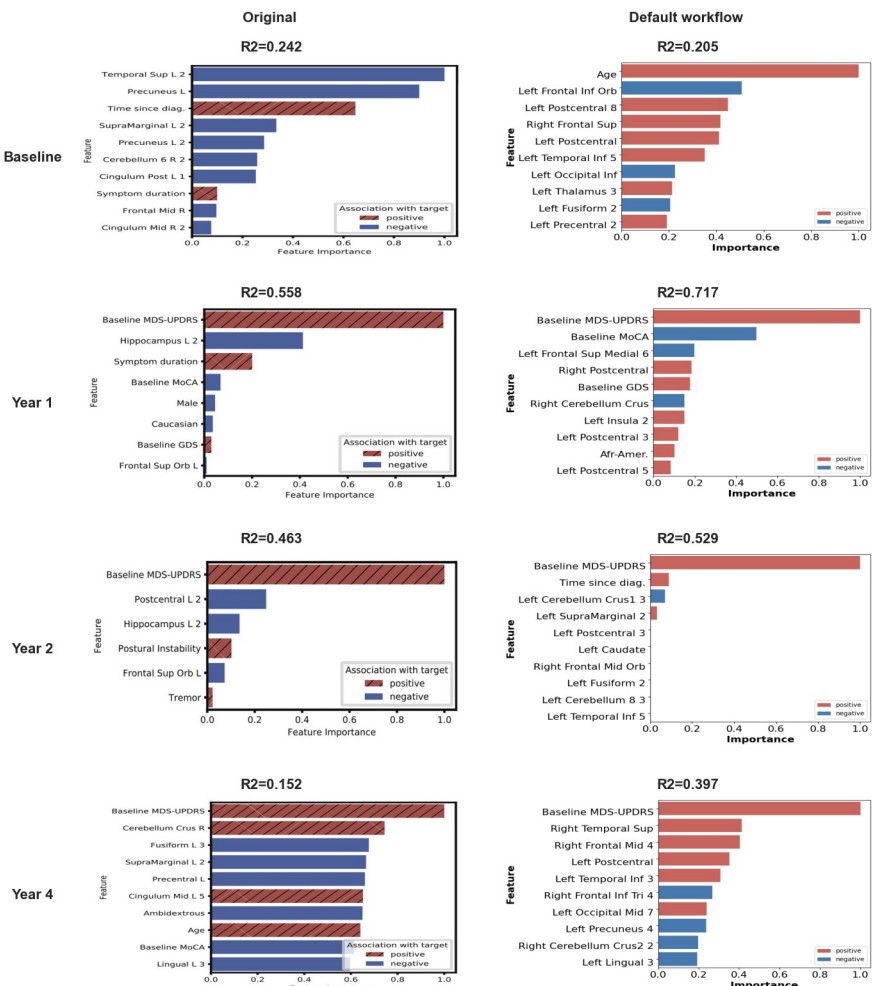

**Fig 6. Predictive features learned by the best performing models to predict MDS-UPDRS score at each time point for the original study (left—extracted from [6]) and the *default workflow* (right) using fALFF.** Features with low importance were not shown. Red bars indicate a positive association and Cyan bars indicate a negative association. Stars (*) represent the presence of this feature in the original study and the default workflow.

found lower performance than the one reported in the original study at Baseline with our *default workflow*. The performance were higher than in the original study at Year 1, Year 2 and Year 4. These values remained close to those reported in the original study and performance were better than chance, supporting the predicting capability of the model reported in the original paper. Thus, using a cohort and methods adapted from [6], we were able to train several machine learning models that predicted Parkinson's disease progression (MDS-UPDRS scores at Baseline, Year 1, Year 2, and Year 4) with a performance higher than chance and with values comparable to those reported in the original study for most models. On these criteria, we could conclude that the replication of the default workflow was successful.

In our partial reproduction attempt, when training the models using a workflow reproducing the authors publicly-available code and the derived data computed by the authors at the time of the original study (fALFF and ReHo whole-brain maps provided at Phase 2), we found close performance to the original ones, except at Year 4 with fALFF, where the default workflow found higher predictability. This confirms the quality of the workflow reproduction for

the clinical and demographics feature selection and for the machine learning part. Note that, only derived rs-fMRI data were provided by the authors, the clinical and demographic features used in this partial reproduction attempt are the ones available in the PPMI database at the time of this study. Thus, potential updates in the PPMI database and on the features might explain the higher predictability observed at that time point during this partial reproduction attempt. Other factors such as differences in scikit-learn version or differences in cross-validation schemes and hyperparameter selections might have impacted the results of the partial reproduction experiments.

Differences in performance with our default workflow could be explained by variations in the pre-processing and imaging features computation pipelines. These could also be explained by differences between cohorts since we had difficulties to exactly reproduce the cohort filtering process of the original study: i.e. our closest-to-original cohort contains, at baseline, 4 participants with different sessions than the original ones, which also impacts follow-up time points cohorts, and potentially the performance of the models. These differences could be related to the evolution of the PPMI database in which sessions were added and removed since the authors queried it for the original study.

In addition, using our default workflow, we found feature importance values that differed —for some predictions— from the ones found by the authors. This is entirely plausible for multivariate machine learning models, and does not preclude the other set of features from not also being useful (e.g. if default gets 0.717, it could be that the features from original are still informative of outcome). This step was complex to replicate since our best performing model x parcellation combination did not match the ones reported in the original paper at several time points, which questions the comparability of the features. When fitting a machine learning model, similar performance can be achieved by different sets of features, which explains why feature importance values might be inconsistent across models.

When introducing specific variations in the workflow, we managed to obtain results that were more similar to the original ones than our default ones, in particular when changing the feature computation method at Baseline. Some changes in the *default workflow* also led to lower performance, for instance at Year 1 and at Year 2 when removing Baseline MDS-UPDRS score or when using only imaging features. For these time points in particular, variations of the pre-processing pipeline (workflows B.), feature computation (workflows C.) and model choice and reporting (workflows E.) had little impact on the performance of the models compared to other time points. We speculate that imaging features were of low importance in the models prediction for these time points compared to other time points (Baseline and Year 4) for which variations on image computation (pre-processing or feature) had a larger impact. Without variations (i.e. with the *default workflow*), performance of models at Baseline and Year 4 time points was already low, which also suggests that effect sizes detected by models were small and that these models were underpowered [19, 51], making them more sensitive to variations. Discussing the predictiveness of the extracted signals for the target outcomes found in the original study is out of the scope of this study. We focus on evaluating the impact of workflow variations in the prediction performance of the models.

In the original study, authors also reported performance of the models evaluated on an external dataset (Table 2 of [6]) and with Leave-One-Site-Out cross-validation (LOSO CV) in the outer-loop compared to Leave-One-Out (LOO CV) in the main study. They found similar performance at Year 1 (R2 over 0.5) with these variations, comparable to the main results in [6] which reported R2 up to 0.558. Performance at other time points was not available for the external validation, but for LOSO CV, models trained for prediction at Year 2 also performed well and those of time point Baseline and Year 4 exhibited lower prediction ability compared to the ones tuned using the LOO CV scheme (main original workflow). This highlights the

importance of model selection and performance reporting, which were also featured prominently in [6]. Some models may have not been optimally tuned, and all models do not have equal capability due to their different functioning, leading to lower performance. The low performance obtained with some models do not put into question the other results, as these have been validated on an external dataset by [6].

When using the replication cohort in which there are differences in the distribution of the most important feature (MDS-UPDRS score at Baseline) of the Year 1 model, a lower performance was found using fALFF ($p < 0.05$) and ReHo. This performance remained high and close to the one reported in the original study. Moreover, when removing specific clinical features such as MDS-UPDRS Baseline scores, the performance models at Year 1 and Year 2 significantly dropped. This suggests that the robustness mentioned above was probably dependant on the distribution of these measures. It would be interesting to assess the interaction of variations in both cohorts, imaging features and input features sets to see if the robustness to analytical variations was also present using the replication cohorts and when increasing the importance of image features in the prediction.

## Challenges of reproducibility studies

In our reproduction attempts, several challenges were encountered, in particular related to cohort selection, fMRI feature pre-processing, and results reporting. To extract the same Baseline cohort as used in [6], we first attempted to query the PPMI database using the information available in the paper and the code publicly available at the start of the reproduction (i.e. without contacting the authors).. This step was unsuccessful since we could not get the same sample size at Baseline (102 instead of 82 in [6]), and we decided to contact the authors who provided us the exact subject and visit list used in the original study. With this list, we were able to build a cohort with the same participants at Baseline. A potential solution to avoid similar difficulties in future reproducibility studies would be to register cohorts obtained from public databases under the same data usage agreements as the original data. In the case of PPMI, a specific section of the online portal could be created to store cohort definitions and associate them with published manuscripts.

Even with the original participant identifiers and visit list at Baseline, we could not retrieve the same Baseline cohort in the PPMI database. Our closest-to-original cohort included the 82 original participants, but for 4 of them, a different visit than the original one was used. For 2 of these visits, we intentionally chose to keep the visits selected by our first query to better fit with the description of the cohort in the paper. For the 2 other visits, the functional images corresponding to these participants and visits were not available anymore in the PPMI database. Since the PPMI database continuously adds new participant visits, we chose to keep only the visits that were added more than a year before the original study publication, since the original authors did not report the date at which they queried the database. With this filter, the Baseline participants list and the exact same code used to search for follow-up visits, the cohorts obtained for follow-up visits were still dissimilar to the original ones, with more participants and several noteworthy differences in clinical and demographic variables. A first step to solve this particular issue would be to systematically report the date when databases are queried. However, the issues faced when attempting to reproduce the original cohort in fact highlight the need for version control in public databases, using tools such as DataLad [52] that is for instance adopted in the OpenNeuro database [53]. With version control, we would be able to retrieve the data from the database as it existed on the date of the original query. In addition, authors would be able to cite the exact version of the database used, which would importantly facilitate cohort reproductions.

Reproducing the fMRI pre-processing and feature computation pipelines described in [6] also raised challenges. First, although authors provided a description of the different pre-processing steps performed and tools used, exact reproductions of neuroimaging pipelines require more detailed information—including specific parameters values, name and version of the standard template used, software versions—given the overall complexity and flexibility of image analysis methods [54]. To build the closest possible pipeline to the one used in [6] without contacting the authors, we had to make informed guesses about important parameters of the analysis. Some of these choices were conditioned by the nature of the neuroimaging pipelines (e.g., the choice of standard template to register functional images was constrained by the use of ICA-AROMA) while other decisions were more arbitrary and led to multiple valid variations (e.g., the computation of WM and CSF mean time-series for which we applied three different variations with different software packages and methods). Reporting guidelines, such as COBIDAS [55], were developed to help document analyses and facilitate reproduction studies. However, to reproduce complete analyses, sharing the entirety of the code used in the original experiment remains the most valuable information, as it contains a both human and machine-readable description of the exact method employed. In our case the authors did provide all code and their custom atlas when asked. Code-sharing platforms such as GitHub and GitLab are now widely available for this purpose and long-term preservation of these code is supported by archive systems such as Software Heritage [56, 57] or Zenodo. We also note that different journals have different requirements regarding what is to be submitted beyond the manuscript. The original paper [6] was published in P&RD which at the time of publication of [6] had minimal expectations beyond the manuscript. The authors met these requirements and beyond, providing a public code repository. Harmonization of such practice across journal would be highly beneficial to help reproduction of studies.

The use of a custom-based atlas to parcellate the brain in the original study also created challenges. Future reproducibility studies would benefit from comprehensive descriptions of the methods used to create such custom data, access to the code to create the data, and sharing of the data itself through platforms such as Zenodo, the Open-Science Framework, Figshare, or NeuroVault [58]. Such platforms could also be used for sharing derived data, for instance whole-brain fALFF and ReHo maps. However, Data Usage Agreements often requires that derived data have to be shared under the same conditions. We emphasize again the need for specific platforms in public databases to host data associated with a published manuscript, including cohort descriptions and derived imaging data.

The authors of [6] shared code used in the original study, in particular for feature computation (fALFF and ReHo after pre-processing and clinical/demographic features search in PPMI study files) and machine-learning models training. The availability of this code was extremely useful for our reproducibility study, and we warmly acknowledge the authors for taking the time to share reusable code with their analysis. Despite the availability of the code, we still faced some difficulties to reproduce the workflow presented in the original study, due to discrepancies between the methods reported in the paper and the code shared, especially for the imaging feature computation, the cross-validation procedure and the results reports. For instance, we were not able to retrieve the Z-scoring of whole-brain fALFF and ReHo maps mentioned in the paper. This discrepancy was likely due to the update of the C-PAC pipeline used by the authors for pre-processing, in which the documentation still mentioned the possibility to output Z-scored maps even if this option was not implemented anymore in the pipeline. This reiterate the importance of code versioning and reporting software versions. The use of software container engines such as Docker and Singularity in combination with frameworks such as Boutiques [37] or BIDS-Apps [59] facilitates reproduction and reduces the technical work required to find and install the software versions used in the original study. The authors

in [6] report that they have begun using both Singularity/Apptainer and Podman for this exact purpose. For more details on the benefits of such software containerization, we refer the readers to [60] in which authors explains how using these particular frameworks can help reproducibility.

Regarding model selection and optimization, we highlight the complexity of nested cross-validation schemes and the on-going debate on the choice of rigorous cross-validation procedures [21, 61]. Here again, code sharing is required to describe the exact evaluation method used in the original study. At this level in the analysis, Jupyter notebooks [62] are an interesting option to document code and mix it with data, natural text and figures. Initiatives were recently launched to share reproducible Jupyter notebooks, such as NeuroLibre [63], a platform for sharing re-executable preprints. We created a Jupyter notebook for our study, that we made publicly available at https://github.com/elodiegermani/nguyen-etal-2021.

The following box highlights the main recommendations that we propose to facilitate the reproduction of such studies in the future.

## Recommendations for more reproducible studies

### *Cohort & Data*

- Creation of specific tools to create and store cohort definitions (participant lists, image IDs, etc.)

- Version control of public databases to be able to access the database as it was at a specific date, and cite the exact version of the database used in the paper

- Share the derived data used in your experiments (using adapted platforms in compliance with regulations wuch as OSF, Figshare, etc.).

### *Pipeline & Code*

- Improve and respect reporting guidelines such as COBIDAS [55]

- Share the entirety of the code on platforms such as GitHub and on platforms for long-term preservation such as Software Heritage [57] or Zenodo

- Harmonization of code and data sharing practice across journals

- Use of containerization tools or containerized software packages to facilitate the retrieval of the exact version used in the study

- For more complex code, use Jupyter Notebooks [62] to facilitate the understandability of the code.

Beyond the limitations related to the challenges of reproducibility, all limitations identified by the authors of [6], including bias of the PPMI cohort towards Caucasian, the use of small sample sizes in particular for prediction at Year 4 and the impact of medication on MDS-UPDRS scores, are also applicable for our study and are further discussed in the original paper [6].

To conclude, we highlighted the challenges associated with the reproduction of neuroimaging studies. We discussed some of the specific difficulties encountered in our study, as well as numerous success in reproduction, and provided some potential solutions to further

facilitate this process in the future, in terms of time cost and adequacy of the reproduction. Nevertheless, given the complexity of the data, software and analyses required in current neuroimaging studies, reproducing the experiments made in existing papers remains extremely challenging.

## Code availability

All the experiments were run using Python 3.10, under a NeuroDocker image available on Dockerhub at https://hub.docker.com/repository/docker/elodiegermani/nguyen-etal-2021. This Docker image contains all necessary package and software used to perform the analysis:

The code used to run the experiments is available in a public notebook on GitHub, and archived in the Software Heritage platform: swh:1:dir:2823c6f1cabae5865aa5ab4d8724e219d5bf2661.

To comply with PPMI's Data Usage Agreements that prevent users to re-publish data, the notebook queries and downloads data directly from PPMI. Since PPMI does not have a data access API, we developed our own Python interface to PPMI using Selenium, a widely-supported Python library to automate web browser navigation. Using this interface, the notebook downloads PPMI study and imaging files to build the cohorts and train the ML models. The utility functions to download and manipulate PPMI data are implemented in LivingPark utils, a Python package available on GitHub (https://github.com/LivingPark-MRI/livingpark-utils).

## Supporting information

**S1 Fig. Comparison of original Schaefer atlas used in [6] and reproduced atlas obtained from three separate atlas available in FSL.**
(TIF)

**S1 Table. Demographics and clinical features set as input for the machine learning models.**
(PDF)

**S2 Table. Terminology of the experiments in the current paper.**
(PDF)

## Author Contributions

**Data curation:** Elodie Germani.

**Formal analysis:** Elodie Germani.

**Investigation:** Elodie Germani.

**Methodology:** Elodie Germani, Nikhil Bhagwat, Mathieu Dugré, Andrzej Sokolowski.

**Resources:** Elodie Germani, Mathieu Dugré.

**Software:** Elodie Germani, Mathieu Dugré, Rémi Gau.

**Supervision:** Madeleine Sharp, Jean-Baptiste Poline, Tristan Glatard.

**Validation:** Elodie Germani, Nikhil Bhagwat, Mathieu Dugré, Andrzej Sokolowski.

**Visualization:** Elodie Germani, Albert A. Montillo, Kevin P. Nguyen.

**Writing – original draft:** Elodie Germani, Albert A. Montillo, Kevin P. Nguyen.

**Writing – review & editing:** Elodie Germani.

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
