## [Decision Letter · Decision Letter 0]

12 Aug 2024

PONE-D-24-21564Predicting Parkinson's disease trajectory using clinical and functional MRI features: a reproduction and replication studyPLOS ONE

Dear Dr. GERMANI,

Thank you for submitting your manuscript to PLOS ONE. After careful consideration, we feel that it has merit but does not fully meet PLOS ONE’s publication criteria as it currently stands. Therefore, we invite you to submit a revised version of the manuscript that addresses the points raised during the review process.

We look forward to receiving your revised manuscript.

Kind regards,

Cota Navin Gupta

Academic Editor

PLOS ONE

www.iitg.ac.in/cngupta

Journal Requirements:

3. Thank you for stating the following financial disclosure: "This work was funded by the Michael J. Fox Foundation for Parkinson's Research (MJFF-021134). This work was also funded by a MITACS Global Research Award (IT34055). This work was partially funded by Region Bretagne (ARED MAPIS) and Agence Nationale pour la Recherche for the programm of doctoral contracts in artificial intelligence (project ANR-20-THIA-0018)."

4. Thank you for stating the following in the Acknowledgments Section of your manuscript: "This work was funded by the Michael J. Fox Foundation for Parkinson’s Research

(MJFF-021134). This work was also funded by a MITACS Global Research Award

(IT34055). This work was partially funded by Region Bretagne (ARED MAPIS) and

Agence Nationale pour la Recherche for the programm of doctoral contracts in artificial

intelligence (project ANR-20-THIA-0018)."

Please remove any funding-related text from the manuscript and let us know how you would like to update your Funding Statement. Currently, your Funding Statement reads as follows: "This work was funded by the Michael J. Fox Foundation for Parkinson's Research (MJFF-021134). This work was also funded by a MITACS Global Research Award (IT34055). This work was partially funded by Region Bretagne (ARED MAPIS) and Agence Nationale pour la Recherche for the programm of doctoral contracts in artificial intelligence (project ANR-20-THIA-0018)."

5. In the online submission form, you indicated that "Data used in the preparation of this article were obtained on August 21st, 2023 from the Parkinson’s Progression Markers Initiative (PPMI) database (www.ppmi-info.org/access-dataspecimens/download-data), RRID:SCR 006431. For up-to-date information on the study, visit www.ppmi-info.org. All data used in this study, as well as a data dictionary, are free and publicly available at the PPMI website, upon an online application, the signature of the Data User Agreement and of the publications policies.

The list of participants and derived data used in this study are available upon request to corresponding authors, upon an online application to the PPMI website, the signature of the Data User Agreement and of the publication policies."

Additional Editor Comments:

Interesting work. However you are requested to respond to queries raised reviewers (especially reviewer 2 and reviewer 4). In revised manuscript please highlight the changes made in a different colour (i.e. for each reviewer) and in reply to reviewers document mention the line numbers of revised manuscript where you have addressed the issue…We look forward to receiving your replies/revisions..Good luck

Reviewers' comments:

Reviewer's Responses to Questions

**Comments to the Author**

1. Is the manuscript technically sound, and do the data support the conclusions?

Reviewer #1: Yes

Reviewer #2: No

Reviewer #3: Yes

Reviewer #4: Partly

2. Has the statistical analysis been performed appropriately and rigorously? 

Reviewer #1: Yes

Reviewer #2: Yes

Reviewer #3: Yes

Reviewer #4: Yes

3. Have the authors made all data underlying the findings in their manuscript fully available?

Reviewer #1: Yes

Reviewer #2: Yes

Reviewer #3: Yes

Reviewer #4: Yes

4. Is the manuscript presented in an intelligible fashion and written in standard English?

Reviewer #1: Yes

Reviewer #2: Yes

Reviewer #3: Yes

Reviewer #4: Yes

5. Review Comments to the Author

Reviewer #1: Review of PONE-D-24-21564: “Predicting Parkinson’s disease trajectory using clinical and functional MRI features: a reproduction and replication study”

Kudos to the authors, for doing a great job of calling attention to the need for reproduction and replication studies. They provide examples of the challenges involved and suggest viable solutions. Reproduction and replication require detailed descriptions of procedures, which the authors elucidate and demonstrate by their example of detailed and meticulous work. The paper reads very well, despite numerous minor English language errors, which should be easy to correct by the English-language editor.

Minor items:

The applications of the terms reproduction and replication are a bit confusing as used in the paper, even though the authors define these terms on p 3, “Here, we define reproducibility as attempts made with the same methods and materials. Replicability, on the other hand, is tested with different but comparable materials or methods, assuming that the tested pipelines are all suitable to extract signal from the data.” Rather than reproduction and replication, strictly speaking the authors demonstrate two flawed attempts at reproduction, one less flawed than the other, arguably approximating a true reproduction. The “reproduction” does not exactly reproduce the original work and the “replication” was not performed on (completely) different data. However, their approach is highly meaningful to help improve both reproduction and replication. Perhaps some more explanatory text could help to avoid confusion and point out that the authors’ use of those terms is somewhat idiosyncratic.

P3. “Reproducibility and replicability of studies in clinical settings is of higher importance to improve the trustworthiness of new biomarkers and to facilitate their development.”--An excellent point. Perhaps this would be beyond the scope of the paper, but it might be worthwhile to also note that confounds such as residual noise (despite nuisance regressors) can affect the utility of the findings (e.g., would be no point in using a scanner as a measure of movement in the scanner if that is what ultimately affects the outcome of the analyses).

P3. “R2 greater than 0 and absolute difference between original and reproduction R2 less than 0.2.”--What led the authors to choose the value 0.2 for this test?

P14. “However, we found higher mean disease durations in the replication cohort than in the original one at all time points, for instance at Baseline with (866.9 days ± 598.7 days) in replication vs (770 days ± 565 days) in original. This difference was not significant at threshold p < 0.05.” Please rethink the approach here and make adjustments. What does p < 0.05 reflect? Surely it does not reflect the probability of having such a large difference after randomly selecting subjects from the same population: The subject selection was not random at all. I think what the authors are trying to say is that in some cases there was nearly no difference in values (as expected for a true reproduction) and in another case the difference seemed meaningful. The authors can either leave this as an intuitive impression or perform some sort of test to illustrate the point, but a different test would be needed.

The figures are blurry. I have had the same problem, which in my case was fixed by changing a parameter in my Windows registry to set the DPI for the image to 300, after which I was able to export publishable figures from Microsoft Office programs.

Reviewer #2: This study explores the replicability of clinical and imaging markers of Parkinson's disease diagnosis and progression prediction, specifically in the context of replicating the work of a previous study in terms of cohort selection, data preprocessing, feature selection/extraction, model selection, and feature importance pipelines. The write-up adequately reflects the undertaken methodical/analytical variations, similarities/differences with replicated work, and findings of the current work. However, some serious study design flaws hinder the scientific utility and impact of this study as noted below.

Foremost, while the write-up is well-detailed in mostly all parts, the structure and flow of the paper are lacking, and the results are not replicated on external data. Specifically, the current flow of the manuscript in several parts is more like that of a rebuttal as compared to a collaborative effort with the authors of the replicated study. This suggests that the collaboration occurred only after a substantial effort had been made to individually replicate the results, thereby limiting the utility of that part of the current work. Moreover, this limitation is unnecessarily amplified by differences in cohort use and even methodological variations between the previous and current studies. Even if it was a late collaboration, I don't see a reason why the exact data and pipeline used in previous work could not be shared between the collaborating teams, thereby eliminating the need to probe a majority of the considered differences and instead focus on enhancing the breadth and potentially the impact of the work (e.g., by exploring additional methods and most importantly replicate qualitative findings of putative biomarkers on additional datasets). As for the difference in the software versions, the previously used pipeline could have been containerized for replication. Lastly, while the central objective is to develop PD progression biomarkers, the qualitative findings of the current study indicate that the derived markers are not really replicable even for same/similar experimentation and within the same dataset. This replication failure (differences in biologically relevant information or the predictivity patterns of the input features) in this collaborative study questions the predictiveness of the extracted signals for the targeted outcomes and/or effectiveness/suitability of the undertaken training/inference approaches.

I hope the authors find my comments to be constructive for current/future work and wish them success in their research.

Reviewer #3: Elodi et al. have replicated and reproduce the study for predicting Parkinsons disease trajectory based on previous study and available information in PPMI database . They have done comprehensive study and have address the problem in reproducing study on predication of parkinson trajectory.

I find the manuscript organised and well written with clear goals. Results are well described and is supported by data. I recommend it for publication.

Reviewer #4: The authors performed an interesting study that tackles an important topic in the attempt to develop neuroimaging biomarkers for clinical implementation in Parkinson’s disease: the robustness of findings. Features are derived from functional MRI and a machine learning model is used to predict disease severity. The methods are thorough and the conclusions are insightful. I do have concerns, however, which largely revolve around the contribution of some of the original authors as co-authors. Their involvement and openness is crucial to performing this study well, but inherently creates a conflict of interest that needs to be addressed more explicitly. Firstly, I urge the authors to add a statement on how neutrality is safeguarded. Secondly, to ensure that the manuscript resonates with all readers and upholds the highest standards of impartiality, I would kindly suggest revisiting the sections where the narrative might seem biased (e.g. remove intensifier words). Further, it is important to clearly define when model performance and feature importance are regarded as different from the original. To do this, it needs to be specified why specific thresholds were used to determine difference between models. Finally, the readability of the manuscript can be improved upon, as a lot of analyses are performed and the reader may get lost in the methods and results. I have provided more details about these and other comments in a point-by-point fashion below.

Detailed points

Abstract

1. “Several neuroimaging biomarkers have been studied recently, but these are susceptible to several sources of variability.”

I think it would help to more concretely define the type of variability of interest for this study. As other types of variability exist as well, but are not covered in this manuscript.

2. I suggest to briefly define replication and reproduction in the abstract.

3. “The success of the reproduction was assessed using different criteria.”

This phrasing is too vague and I suggest to rephrase it.

4. Only the replication results are mentioned and not the reproduction results. To accurately reflect the contests of the manuscript, those results should be summarized here as well.

Introduction

5. The first paragraph mentions “disease subtypes” on various occasions, but this is not studied in the current manuscript. It may be more clear to only mention stages, severity or progression, as that aligns more closely with the manuscript.

6. “While disease phenotypes are heterogeneous, neuronal dysfunction patterns were shown to be highly replicable between patients.”

I would like to see the latter statement explained more in depth. It also seems to be phrased a bit too strongly, also considering the explained variance in the final models of this study. These patterns can still be important, but I think more contextualization is important.

7. Why may the ALFF and ReHo measures be particularly relevant? This needs to be apparent for the reader without requiring them to read the original manuscript.

8. “This suggests that a single pipeline evaluation is not sufficient to obtain robust results, though the reliability of results may be increased when studying their distributions across perturbations.”

The second part of this sentence may be a bit difficult to follow for non-expert readers. I suggest to give a little more context

9. “Note that comparable is ambiguous, but defined further in this case in Method Section.”

It may be helpful to at least briefly summarize the definition here. Also, the second part of the sentence does not read very well. I suggest rephrasing it slightly.

10. “Replicability experiments have shown different degrees of variability between findings obtained with different analytic conditions. These studies are usually done using healthy populations and in general research practice (as opposed to clinical research), as in [16]. For clinically-oriented research, however, the topic remains understudied.”

The difference between clinically-oriented research and general research practice needs to be defined more clearly.

11. “Reproducibility and replicability of studies in clinical settings is of higher importance to improve the trustworthiness of new biomarkers and to facilitate their development”

Until now the introduction mostly covers the robustness and variability of biomarkers, which clearly links to the trustworthiness of the findings. However, I suggest to provide more details on how this facilitates the development of them.

12. “… a clinically-oriented research …”

This seems grammatically incorrect. I suggest to change “research” to “study”.

13. The involvement of authors of the original paper (e.g. KPN and AAM) is crucial to make accurate comparisons to that work. Their openness to share resources, as well as being involved in the study despite some slightly conflicting results being reported is commendable. Unfortunately, even with the best intentions, their direct involvement as co-authors does make it difficult to judge the independence of the statements made in the manuscript. For transparency, I think it is important to add a statement mentioning how this was handled.

14. The abstract mentions that this study is part of a larger effort to replicate imaging biomarkers in PD. This should be mentioned in the introduction as well. On top of that, given the context of this broader effort, I suggest to specify what sets this specific study apart.

Methods

15. The terms reproduction and replication have been well defined in the introduction. However, throughout the manuscript they seem to not have been used fully consistently. For instance, the first paragraph of the methods contains the phrase “This two-step reproduction was meant…”. However, reproduction here refers to both reproduction and replication. I recommend thoroughly checking this throughout the manuscript to avoid confusion.

16. The choice to start with a replication step without contacting the authors should be motivated in more detail in the introduction. Currently it is not made clear enough as to why this would be an important first step.

17. Relatedly, for workflows as well as cohorts, it was hard to keep track which related to replication and which to reproduction. This needs to be made more clear. A suggestion would be to consistently use the words “replication” and “reproduction” when naming the workflows and cohorts.

18. The “default workflow” was defined as the workflow that was “most likely” according to the code shared along with the paper. It is not fully clear wat “most likely” means here. This is explained more thoroughly in the paragraphs below, but it may help to briefly inform the reader that details will be given below.

19. The scanning acquisition parameters need to be included.

20. The authors chose to filter the database and keep only the participants for which the visit/archive date was before January 1st, 2020. I am wondering why a date was used an not use the number of participants for this? Selecting a certain date feels more arbitrary.

21. A few variations of the original pipeline were tested to see how these would affect the results. It is not fully clear, however, why these particular variations were chosen. I would like to ask the authors to elaborate on this.

22. “We note that for the second step of the reproduction experiment, the authors of [1] have supplied us with all derived maps.”

Here, it is not completely obvious what “second step” refers to.

23. The discussion about the atlases in the “imaging features computation” section should be in a distinct subsection.

24. “… they had downloaded, through they did factor prominently into their results, in order to understand better the relevance of the database update.”

The final part of this sentence does not seem grammatically correct. There is also a typo here: through  though

25. In the section on “Evaluation metrics”, I suggest to also explain what the RMSE represents, just like was done for R2.

26. Null performance was tested using the ElasticNet and Schaefer atlas, but it is not explained why these were chosen. Please elaborate on this.

27. Was the evaluation of the models’ ability to classify high versus low severity patients also evaluated in the original study? This is not fully clear to me from the current text.

28. It needs to be defined when model performance will be regarded as different from the original and when they will be regarded as similar. The same is true for feature importance.

Results

29. Throughout the results section, statistical parameters either seem to be missing or simply limits being reported (e.g. <0.05). These parameters need to be reported more comprehensively and with exact values.

30. When differences were observed between cohorts, I suggest to specify the values of each cohort, similarly to what was done for disease duration in paragraph 3.

31. “Most participants of the study showed high movement parameters. Indeed, out of 102, 80 showed at least one time point with a frame-wise displacement superior to 0.5mm.”

What was the mean frame-wise displacement?

32. Sometimes the R2 is reported as fraction and sometimes as percentage. This needs to be the same throughout the results.

33. The results section contains too much discussion and interpretation that are repeated in the discussion. Readability of the manuscript would be improved if this section is written more concisely.

34. “In Fig 4, we can see that using authors derivatives and thus, the original cohort, we achieve performance that are very close to the original ones, except at Year 4. This informs us on the quality of the reproduction of the clinical and demographic features selection, but also on the machine learning models training and selection”

The observation that the performance is not the same for year 4 should be addressed. If this is not close to the original values, then something must be different.

35. In the “robustness to workflow variations” section, “models performance” is grammatically incorrect.

36. Typo: “…values with for fALFF..”  for

37. “Feature importance showed relatively few overlap between the ones obtained using our models and those reported in the original study, especially for imaging features, at all time points”

This is an important point that needs to be covered more extensively in the discussion, as it calls the reliability of the imaging measures into question.

Discussion

38. “When training the models using the derived data computed by the authors at the time of the original study (fALFF and ReHo whole-brain maps), we found very close performance to the original ones, except at Year 4 with fALFF, where the default worklow found even higher predictability. This confirms the quality of the reproduction for the clinical and demographics feature selection and for the machine learning part.”

A higher predictability at Year 4 with fALFF still is a different performance. This was a recurrent statement throughout the discussion that needs to be addressed accordingly.

39. In that same sentence there is a typo: worklow  workflow

40. “When using a different cohort with distinctions in the distribution of the most important feature (MDS-UPDRS score at Baseline) of the Year 1 model”

Given all the workflow and cohorts that the reader needs to track, I suggest being more specific than “a different cohort” and use names of cohorts consistently.

41. “It would be interesting to assess the interaction of variations in both cohorts, imaging features and input features sets to see if the robustness to analytical variations was also present using the replication cohorts and when increasing the importance of image features in the prediction.”

What answers could this give exactly? And, if interesting, why not do it?

42. “…we first attempted to query the PPMI database using the information available in the paper and the code shared by the authors.”

I believe this refers to the publicly available code, but since the original authors were included later and shared resources at that point as well this may be confusing. I suggest clarifying this.

43. In the discussion it was noted that five participants had a different visit in the closest-to-original cohort, but the results only mentions four. Please check this.

44. The discussion describes that select software could help with replicability, but it is still unclear which specific software tackles what specific challenges. I would like to see more detail on this.

45. Relatedly, it would be helpful to add a box/figure with very specific point-by-point recommendations for future studies. I think this could facilitate adoption of your recommendations.

46. The discussion contains several intensifiers that add subjective emphasis (e.g. “slightly” or “very”) when describing differences to the original study. These need to be used more carefully.

47. Relatedly, the last sentence notes that “reproducing existing papers remains extremely challenging”, whilst the discussion often notes how the reproduced results were close to the original. This discrepancy is confusing and should be addressed.

48. Some additional limitations need to be mentioned, such as those mentioned in the original manuscript.

Figures

49. Figure 1 is referenced as “Figure 1” on one occasion, whereas all other references to the figures use “Fig” in the main text. I suggest to align this.

50. Figure 1. It would be helpful if here the authors could visualize which workflows count as “replication” and which as “reproduction”.

51. Figure 3. The binning of the replication cohort at year 1 seem different from the closest-to-original and original cohorts. This may need to be adjusted for accurate comparisons.

52. Figure 3. Why were Year 2 and 4 not reported?

53. Figure 6. I suggest to use the same coloring scheme for easier visual comparisons of the two atlases.

6. PLOS authors have the option to publish the peer review history of their article (what does this mean?). If published, this will include your full peer review and any attached files.

Reviewer #1: No

Reviewer #2: No

Reviewer #3: **Yes: **Rakesh Kumar

Reviewer #4: **Yes: **Tommy A.A. Broeders

---

## [Author Response · Author response to Decision Letter 0]

10 Nov 2024

We prepared the revised version of the manuscript taking into account the journal requirements and the suggestion of the reviewers, in particular:

• We modified the style of the manuscript to meet PLOS ONE's style requirements.

• As suggested by the Editor, we would like to add the following statement regarding the role of funders: “The funders had no role in study design, data collection and analysis, decision to publish, or preparation of the manuscript.”. We also removed funding-related texts from the revised manuscript.

• Concerning the availability of derived data, publicly sharing the participant list and derived data used in this study would breach the compliance to PPMI Data Use Agreement, which is why we made these data available upon request.

We also made some modifications according to the reviewers comments, in particular:

• We clarified the two steps of our study to better explain the involvement of the authors and to avoid any confusion between reproduction and replications, as suggested by Reviewers #1 and #4:

• Phase 1 corresponding to the first reproduction attempt without contacting the authors.

• Phase 2 corresponding to the second attempt after contacting the authors and using

derived data that they provided to us.

• We added some clarifications, in particular on the use of specific thresholds to compare results (Reviewer #1 and #4). We removed statistical testings performed between cohorts as suggested by Reviewer #1. We thus modified the paragraph related to these results and tried to improve its readability as suggested by Reviewer #4.

• We also added a summary of the main recommendations discussed in the paper to improve the reproducibility of future studies.

A complete point-by-point reply to the reviewer's comments is provided along with this new revision. We hope that these modifications will help improve the quality of our final manuscript.

---

## [Decision Letter · Decision Letter 1]

10 Dec 2024

PONE-D-24-21564R1Predicting Parkinson's disease trajectory using clinical and functional MRI features: a reproduction and replication studyPLOS ONE

Dear Dr. GERMANI,

Thank you for submitting your manuscript to PLOS ONE. After careful consideration, we feel that it has merit but does not fully meet PLOS ONE’s publication criteria as it currently stands. Therefore, we invite you to submit a revised version of the manuscript that addresses the points raised during the review process.

We look forward to receiving your revised manuscript.

Kind regards,

Cota Navin Gupta

Academic Editor

PLOS ONE

www.iitg.ac.in/cngupta

**Journal Requirements:**

**Additional Editor Comments:**

The manuscript is much improved however the authors are requested to address the MINOR COMMENTS raised by reviewers. Please follow below instructions while addressing the raised queries.

In revised manuscript please highlight the changes made in a different colour (i.e. for each reviewer) and in reply to reviewers document mention the line numbers of revised manuscript where you have addressed the issue…We look forward to receiving your replies/revisions..Good luck.

Reviewers' comments:

Reviewer's Responses to Questions

**Comments to the Author**

1. If the authors have adequately addressed your comments raised in a previous round of review and you feel that this manuscript is now acceptable for publication, you may indicate that here to bypass the “Comments to the Author” section, enter your conflict of interest statement in the “Confidential to Editor” section, and submit your "Accept" recommendation.

Reviewer #1: (No Response)

Reviewer #4: (No Response)

2. Is the manuscript technically sound, and do the data support the conclusions?

Reviewer #1: Yes

Reviewer #4: Yes

3. Has the statistical analysis been performed appropriately and rigorously? 

Reviewer #1: Yes

Reviewer #4: Yes

4. Have the authors made all data underlying the findings in their manuscript fully available?

Reviewer #1: Yes

Reviewer #4: Yes

5. Is the manuscript presented in an intelligible fashion and written in standard English?

Reviewer #1: Yes

Reviewer #4: Yes

6. Review Comments to the Author

**Reviewer #1: **Review of PONE-D-24-21564_r1: “Predicting Parkinson’s disease trajectory using clinical and functional MRI features: a reproduction and replication study”

The authors have done a fine job of meticulously addressing reviewer comments. However, two of my comments have not been fully addressed:

The language concerning the terms reproduction and replication has been much improved and far less confusing, but there are still some points of potential confusion. The authors seem to contradict themselves at times. For example, in the abstract “Here, we attempt to reproduce (re-implementing the experiments with the same data, same method) …” defines a reproduction as involving not only the same methods, but also the same DATA. In the abstract we also find “Moreover, using derived data provided by the authors of the original study, we were able to make an EXACT reproduction and managed to obtain results that were close to the original ones.” This begs the question, if the same data and same methods were used, why aren’t the results the same? I don’t find this question answered. From lines 601-608 we find “The closest-to-original cohort exhibited almost the same characteristics as the original one at Baseline. For subsequent time points, we found some differences, in particular at Year 2 and at Year 4: participants were older in the closest-to-original cohort than in the original study at Year 4 (63.8 ± 11.0 in the closest to original cohort compared to 62.1 ± 9.8 in the original), Baseline mean MDS-UPDRS score was lower for Year 2 (40.2 ± 18.2 in original, 35.2 ± 16.1 in closest-to-original) and Year 4 (34.9 ± 15.7 in original, 26.1 ± 11.4 in closest-to-original) and mean MDS-UPDRS score at prediction time point was similar to the original cohort except at Year 4.” And on lines 902-905 “Even with the original participant identifiers and visit list at Baseline, we could not retrieve the same Baseline cohort in the PPMI database. Our closest-to-original cohort included the 82 original participants, but for 4 of them, a different visit than the original one was used.” Are the authors actually dealing with the EXACT same data in their reproduction? Perhaps the authors could avoid confusion by qualifying that the data are nearly exactly the same and/or that the reproduction was attempted rather than an EXACT reproduction.

The figures are still blurry, but this issue can wait until final preparation for publication.

**Reviewer #4: **I would like to thank the authors for addressing all my comments. All my concerns have been resolved. I have one final comment regarding the removal of p-values, which I agree with as the samples have not been gathered randomly and independently. However, a definition of when values are considered "different" is missing and I suggest specifying this.

7. PLOS authors have the option to publish the peer review history of their article (what does this mean?). If published, this will include your full peer review and any attached files.

Reviewer #1: No

Reviewer #4: **Yes: **Tommy A.A. Broeders

---

## [Author Response · Author response to Decision Letter 1]

14 Dec 2024

We would like to thank the reviewers for their thoughtful comments, and for recognizing our efforts. To answer their remaining comments, we made some modifications to the manuscript to clearly distinguish between the use of same data (derived data provided by the authors) and the use of the same data collection method (using the list of participants and sessions provided by the authors). We also emphasize the potential reasons for differences in the results using same data. We also added a specification criteria to determine the "meaningful" differences between cohort demographics. We hope these changes will answer your comments and thank you again for reviewing our manuscript.

---

## [Decision Letter · Decision Letter 2]

2 Jan 2025

Predicting Parkinson's disease trajectory using clinical and functional MRI features: a reproduction and replication study

PONE-D-24-21564R2

Dear Dr. GERMANI,

We’re pleased to inform you that your manuscript has been judged scientifically suitable for publication and will be formally accepted for publication once it meets all outstanding technical requirements.

Kind regards,

Cota Navin Gupta

Academic Editor

PLOS ONE

Additional Editor Comments (optional):

Thanks for addressing concerns raised by me and reviewers....Congrats on the work done

Reviewers' comments:

Reviewer's Responses to Questions

**Comments to the Author**

1. If the authors have adequately addressed your comments raised in a previous round of review and you feel that this manuscript is now acceptable for publication, you may indicate that here to bypass the “Comments to the Author” section, enter your conflict of interest statement in the “Confidential to Editor” section, and submit your "Accept" recommendation.

Reviewer #1: All comments have been addressed

Reviewer #4: All comments have been addressed

2. Is the manuscript technically sound, and do the data support the conclusions?

Reviewer #1: (No Response)

Reviewer #4: (No Response)

3. Has the statistical analysis been performed appropriately and rigorously? 

Reviewer #1: (No Response)

Reviewer #4: (No Response)

4. Have the authors made all data underlying the findings in their manuscript fully available?

Reviewer #1: (No Response)

Reviewer #4: (No Response)

5. Is the manuscript presented in an intelligible fashion and written in standard English?

Reviewer #1: (No Response)

Reviewer #4: (No Response)

6. Review Comments to the Author

Reviewer #1: (No Response)

Reviewer #4: (No Response)

7. PLOS authors have the option to publish the peer review history of their article (what does this mean?). If published, this will include your full peer review and any attached files.

Reviewer #1: No

Reviewer #4: **Yes: **Tommy A.A. Broeders

---

## [Editor Report · Acceptance letter]

7 Jan 2025

PONE-D-24-21564R2 

PLOS ONE

Dear Dr. GERMANI, 

I'm pleased to inform you that your manuscript has been deemed suitable for publication in PLOS ONE. Congratulations! Your manuscript is now being handed over to our production team.

Kind regards, 

on behalf of

Dr. Cota Navin Gupta 

Academic Editor

PLOS ONE